# MM-Spectrum: Multimodal Multi-spectral Molecular Structural Elucidation with a Stable MoE Framework

Hai-tao Yu [1]  Nan Min [2]  Zheng Fang [1]  Hongyu Zhan [1]  Yusen Tan [1]  Yuhan Wang [3]  Jun Xia [1 4]

## Abstract

Inferring molecular structures from multimodal spectroscopic measurements requires integrating complementary yet highly heterogeneous signals. However, the common paradigm of directly concatenating multispectral sequences can exhibit anomalous performance degradation, primarily due to pronounced heterogeneity and the resulting multimodal imbalance across modalities. As a remedy, we propose **MM-Spectrum**, a sparse Mixture-of-Experts framework tailored for multimodal multispectral spectra-to-structure elucidation. To better match the information characteristics under multispectral imbalance, MM-Spectrum introduces an explicit modality-aware routing mechanism that exposes spectral identity to the router in addition to token content representations. Moreover, it incorporates *shared* and *interaction* experts, together with heterogeneous expert capacities, to extract multispectral modality-unique and cross-modal synergistic information while suppressing noise-induced interference. Across full-modality, bimodal, and missing-modality settings on molecular structural elucidation, MM-Spectrum achieves consistent and substantial improvements, supported by ablation studies and interpretability analyses. Code is available at https://github.com/HHHTTY/MM-Spectrum.

## 1. Introduction

Spectroscopy-driven *molecular structural elucidation* is a foundational capability for chemical discovery (Alberts et al., 2024). In learning-based elucidation, the key challenge is often not whether a *single* modality can predict

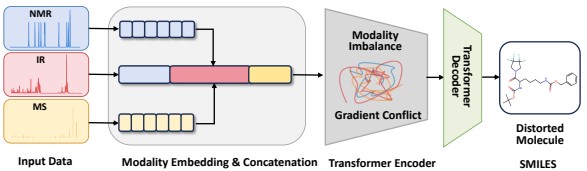

(a) **Mechanism of Failure:** the naive concatenation baseline leading to MultiSpectral Imbalance.

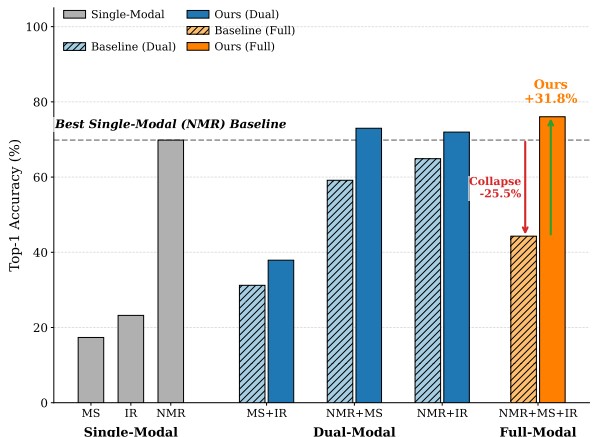

(b) **Empirical Consequence:** Full-modality inputs cause a performance "collapse" in the naive concatenation baseline, while MM-Spectrum achieves synergy.

*Figure 1.* The Challenge of Multispectral Elucidation: Mechanism and Consequence. In contrast, MM-Spectrum resolves this imbalance, effectively utilizing all modalities to achieve state-of-the-art performance.

molecular structures, but whether *multiple* modalities can achieve *stable synergy* (Liang et al., 2024). Unlike generic multimodal learning settings, spectroscopic modalities exhibit complementarity with clear physical meaning. Nuclear magnetic resonance (**NMR**) provides high-resolution constraints on local atomic environments and molecular topology, yet its nonlinear peak patterns lead to complex symbolic structures (Klukowski et al., 2025). Infrared spectroscopy (**IR**) offers coarse evidence of functional groups, but is typically noisy and low in information density, requiring long fragments to form stable semantics (Yang et al., 2021). Mass spectrometry (**MS**) contributes strong global constraints through molecular mass and fragmentation, but

[1]The Hong Kong University of Science and Technology (Guangzhou) [2]Southeast University [3]Fudan University [4]The Hong Kong University of Science and Technology. Correspondence to: Jun Xia <junxia@hkust-gz.edu.cn>.

*Proceedings of the 43rd International Conference on Machine Learning*, Seoul, South Korea. PMLR 306, 2026. Copyright 2026 by the author(s).

is limited in distinguishing many stereoisomers and constitutional isomers (Dührkop et al., 2015).

In molecular structural elucidation, multispectral sequences are concatenated into a long input and mapped to molecular structures in an end-to-end manner. We focus on a counterintuitive phenomenon in multispectral modeling: providing full-modality inputs does not necessarily outperform unimodal settings and can even lead to substantial degradation. As illustrated in Figure 1, naive full-modality concatenation can underperform strong unimodal or bimodal baselines, and the full-modality model may drop sharply compared to the NMR-only setting. We attribute this failure primarily to the heterogeneity and imbalance of multispectral data (Wu et al., 2022). Specifically, modality distributions differ drastically, and simple concatenation makes "long but weak" modalities dominate attention: IR sequences are often extremely long with strong token autocorrelation, whereas NMR is shorter but has much higher semantic density. Meanwhile, modalities also differ structurally: NMR often contains structured symbols, while MS and IR are closer to peak lists or locally correlated continuous statistics. During optimization, different modalities can induce competing gradient directions. Low-density modalities may dominate the update process, and such gradient dominance can force the model toward a suboptimal compromise among competing objectives (Yu et al., 2020).

To address these issues, we reconstruct multispectral fusion from naive concatenation into a structured system of expert division of labor. Multispectral fusion should not be treated as a crude additive aggregation of evidence, but rather as a structured allocation and coordination process over heterogeneous sources. Mixture-of-Experts (**MoE**) provides a natural path: via sparse activation, each modality invokes only a small subset of experts, increasing effective capacity while enabling specialization. However, the balancing assumptions in standard MoE do not automatically match the severe imbalance in multispectral elucidation, and MoE can introduce new instabilities (Fedus et al., 2022). We therefore propose **MM-Spectrum**, a stable MoE framework designed specifically for multispectral imbalance and information synergy. We view multispectral fusion as a problem of *structured allocation and information decomposition under imbalance constraints*. To better fit the imbalanced characteristics of multispectral signals, we design an explicit modality-aware routing mechanism, allowing the router to access not only token content representations but also explicit spectral modality identities, which reduces routing difficulty and stabilizes specialization (Zhou et al., 2022).

Moreover, multispectral modalities are not simply complementary, they involve both redundancy and synergy (Williams & Beer, 2010). This property also explains why naive concatenation fails, as weaker modalities may be un-

able to contribute effectively to molecular structural elucidation. MM-Spectrum introduces shared experts and interaction experts to capture cross-modality redundancy and synergy, and further incorporates heterogeneous expert capacities with computation-cost regularization (Du et al., 2022). Under multispectral inputs, this design encourages the model to extract modality-unique and synergistic information while suppressing interference from noise. As a result, MM-Spectrum not only improves performance but also provides interpretable evidence for mitigating multispectral imbalance, supporting a synergistic learning mechanism for multispectral complementarity.

**Contributions.**  MM-Spectrum makes the following contributions:

- We are the first to identify multimodal imbalance induced by heterogeneity in multispectral spectra-to-structure elucidation, and to interpret it through information-density disparity and gradient conflicts.

- We propose **MM-Spectrum**, a spectroscopy-oriented stable sparse MoE framework with modality-aware routing and a structured expert space that separates redundant, unique, and synergistic information pathways. We further introduce heterogeneous experts with computation-cost regularization to improve accuracy while reducing training and inference overhead.

- We demonstrate consistent improvements in full-modality, bimodal, and missing-modality settings, supported by ablation studies and interpretability analyses.

## 2. Related Work

### 2.1. Spectra-to-Structure Elucidation

Computational molecular structural elucidation has evolved from database-driven search to generative modeling. Classical approaches typically rely on retrieving candidates from large molecular databases using spectral fingerprints. Representative systems such as CSI:FingerID (Dührkop et al., 2015) and SIRIUS 4 (Dührkop et al.) utilize kernel support vector machines to predict molecular fingerprints from tandem mass spectrometry (MS/MS) data. Complementary to MS, NMR-based elucidation has traditionally depended on axiomatic construction algorithms or database lookups to assemble structures from chemical shift constraints (Klukowski et al., 2025).

Recently, deep generative models have been increasingly applied to this task. For mass spectrometry, autoregressive models such as MassGenie (Shrivastava et al., 2021) and Spec2Mol (Litsa et al., 2023) generate molecular structures from mass spectra, while diffusion-based approaches such as DiffMS (Bohde et al., 2025) explore alternative

conditional generation mechanisms. Similarly, for NMR, approaches range from token-based generation (Klukowski et al., 2025) and multitask frameworks like NMR2Struct (Hu et al., 2024) to graph neural networks that reconstruct atomic environments from chemical shifts (Guan et al., 2021). Furthermore, for IR spectroscopy, contrastive learning frameworks such as Spectra-to-Structure (Kanakala et al., 2024) have been explored. In the multimodal regime, however, the prevailing baseline remains naive early-fusion, where heterogeneous modalities are concatenated or pooled before processing (Yang et al., 2021), although some recent works explore contrastive alignment to improve cross-modal consistency (Liang et al., 2024). While architecturally simple, the concatenation paradigm implicitly assumes that token semantics and information densities are comparable across modalities. Recent studies in multimodal learning suggest this assumption is often violated: modalities with different convergence rates or noise levels can lead to *negative transfer* or greedy optimization behavior, where a dominant modality suppresses the learning of others due to gradient conflicts (Wang et al., 2020; Yu et al., 2020; Wu et al., 2022).

### 2.2. Mixture-of-Experts and Balancing Mechanisms

Sparse Mixture-of-Experts (MoE) models scale model capacity without a proportional increase in computational cost by conditionally activating a subset of parameters. The sparsely-gated MoE layer (Shazeer et al., 2017) introduced a learnable gating network to route tokens to top-$k$ experts, a design later scaled to trillion-parameter regimes by GShard (Lepikhin et al., 2021) and Switch Transformers (Fedus et al., 2022). To prevent "expert collapse"—where the router trivially assigns all tokens to a single expert—these models universally employ auxiliary balancing losses based on expert importance and load statistics (Fedus et al., 2022; Du et al., 2022). Such conditional computation has also been successfully adapted to other domains, such as vision, to handle patch-level redundancy (Riquelme et al., 2021).

However, standard balancing formulations are predicated on the assumption that tokens are relatively homogeneous in information content and should be distributed uniformly. Strict uniform balancing often hinders specialization by misallocating high-capacity experts to low-utility noise tokens(Zhou et al., 2022; Lewis et al., 2021). MM-Spectrum introduces a spectrum-specific inductive bias, leveraging modality-aware routing and cost-regularization to align expert specialization with the physical nature of spectrum.

## 3. Methodology

### 3.1. Preliminaries

**Task Definition**   We consider multimodal spectroscopy-based *Molecular Structural Elucidation* with $M$ modalities (in this work $M = 3$, corresponding to NMR, IR, and MS). For modality $m \in \{1, \ldots, M\}$, the input is a token sequence:

$$x^{(m)} = \left(x_1^{(m)}, x_2^{(m)}, \ldots, x_{L_m}^{(m)}\right). \quad (1)$$

The multimodal observation is denoted by $X = \{x^{(m)}\}_{m=1}^M$, and the target molecular structure is represented as a discrete sequence $y = (y_1, \ldots, y_T) \in \mathcal{Y}$ (e.g., a SMILES string). Using an encoder–decoder Transformer, the encoder produces contextual representations and the decoder generates $y$ autoregressively. The complete autoregressive likelihood formulation and training objective are provided in Section A.1 of the appendix.

**Standard Sparse Mixture-of-Experts**   To scale model capacity without proportional computational costs, Sparse Mixture-of-Experts (MoE) replaces the dense feed-forward network (FFN) with a set of $E$ experts $\{\text{FFN}_e\}_{e=1}^E$. For an input token representation $h$, a learnable router $W_r$ computes a probability distribution $p(h)$ and activates only the top-$k$ experts. While efficient, standard MoE routing relies solely on token content $h$. In multimodal spectroscopy, this design is prone to failure due to extreme sequence imbalance: abundant, low-density tokens (long IR sequences) often numerically dominate the routing statistics, causing the model to neglect sparse but constraint-rich signals (NMR). We address this limitation in **MM-Spectrum** by introducing modality-aware routing and structured expert subspaces.

### 3.2. Spectrum-Aware Representation

As illustrated in Figure 2, MM-Spectrum restructures the spectra-to-structure generation pipeline to address the intrinsic statistical imbalance where long, low-density modalities (MS and IR) numerically dominate short, high-density constraints (NMR). The framework integrates three design principles: (i) aligning token statistics via compression, (ii) exposing modality identity for routing, and (iii) decoupling information flow via structured expert subspaces.

To mitigate the mismatch between sequence length $L_m$ and semantic density $\rho_m$, we formalize data preprocessing as modality-specific operators $\phi_m$ that map raw inputs $x^{(m)}$ to compressed representations $\tilde{x}^{(m)}$:

$$\tilde{x}^{(m)} = \phi_m(x^{(m)}), \qquad \tilde{X} = \{\tilde{x}^{(m)}\}_{m=1}^M. \quad (2)$$

We instantiate $\phi_m$ to balance information retention against computational cost based on signal topology. For **High-Density Modalities** (NMR), we set $\phi_{\text{NMR}} \approx$ Identity to preserve discrete, constraint-rich topology signals without smoothing. Conversely, for **High-Redundancy Modalities** (IR/MS), we apply adaptive compression—specifically local binning for autocorrelated IR and top-$k$ filtering for noisy MS—to excise background noise. This preprocessing

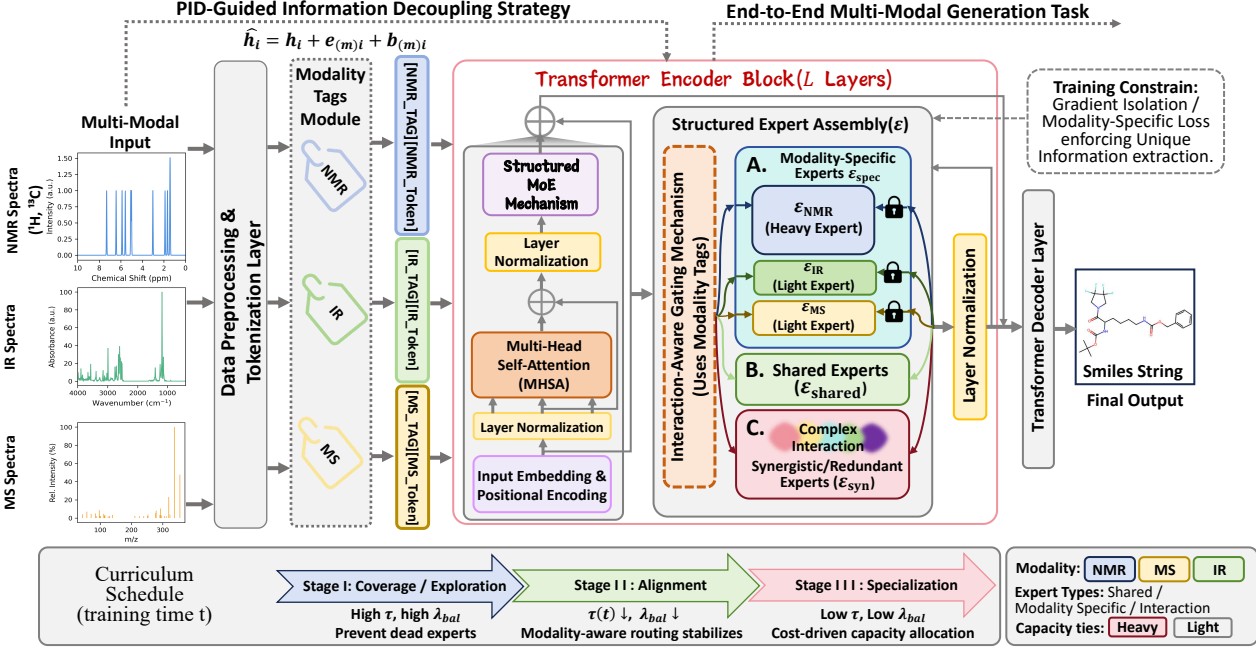

*Figure 2.* **MM-Spectrum overview: PID-guided information decoupling with structured MoE and curriculum scheduling.** We formulate multimodal spectra-to-structure generation as an end-to-end encoder–decoder task, where each modality is tokenized with explicit *Modality Tags*. The encoder block follows LayerNorm → MHSA → *Structured MoE* → LayerNorm, where an *interaction-aware gating mechanism* leverages modality tags to route tokens into a *structured expert assembly*.

effectively aligns the effective density $\rho_m$ across modalities, preventing attention dilution and ensuring stable gradient statistics for the subsequent MoE layers. Detailed modality-specific tokenization, compression operators, and sequence configurations are provided in Section B.1 of the appendix.

### 3.3. Spectrum-Aware Routing

Standard content-based routing requires the gate to implicitly infer signal origin from noisy token statistics, which is suboptimal for heterogeneous spectroscopy. MM-Spectrum explicitly injects modality information into the routing latent space, decoupling *identity* from *content* (see Figure 4). For a token $i$ with content representation $h_i \in \mathbb{R}^d$ and modality index $m(i)$, we introduce a learnable tag embedding $e_{m(i)}$ (Vaswani et al., 2017) and a modality bias $b_{m(i)}$ to construct the router input $\tilde{h}_i$:

$$\tilde{h}_i = h_i + e_{m(i)} + b_{m(i)}. \tag{3}$$

The gating network then computes expert probabilities and performs Top-$k$ selection:

$$
\begin{aligned}
p_{i,e} &= \mathrm{softmax}\left(\frac{w_e^\top \tilde{h}_i}{\tau}\right)_e, \\
o_i &= \sum_{e \in \mathrm{TopK}(p_{i,:},k)} p_{i,e} \cdot \mathrm{FFN}_e(h_i).
\end{aligned}
\tag{4}
$$

Geometrically, Eq. (3) induces a translation-like bias in the router's decision space. The term $b_{m(i)}$ encodes a disentangled "modality preference," stabilizing early-stage coarse differentiation while preserving the metric space of $h_i$ for fine-grained content matching.

Crucially, this mechanism promotes **Soft Specialization**. Unlike hard-coding specific experts to modalities (which causes gradient blocking and limits synergy), MM-Spectrum allows specialization to *emerge* dynamically. Guided by the explicit cues in $\tilde{h}_i$ and constrained by computation-aware regularization, the router learns to direct tokens to appropriate experts—allocating unique constraints to specific experts and cross-modal patterns to interaction experts—without forfeiting the flexibility to exploit shared parameters when beneficial. Additional derivations of the modality-conditioned routing formulation and the associated stability-to-specialization schedule are given in Sections A.4 and A.6 of the appendix.

### 3.4. Structured Expert Space

To effectively disentangle heterogeneous signals, we adopt Partial Information Decomposition (PID) (Williams & Beer, 2010) as a theoretical inductive bias. As conceptualized in Figure 3, we posit that the mutual information between multimodal spectra $X$ and structure $y$ decomposes into re-

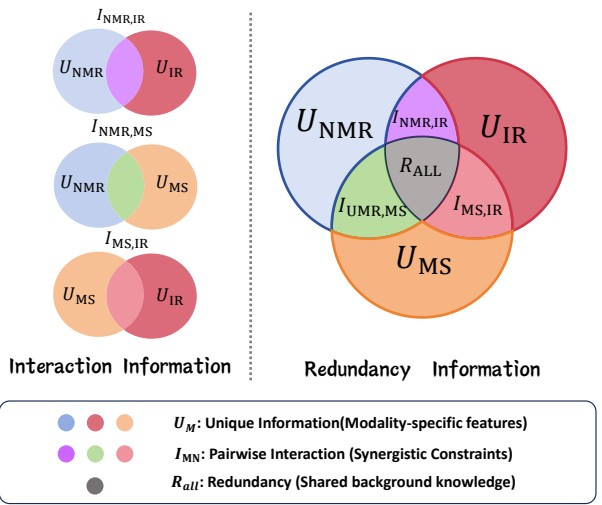

*Figure 3.* **Decomposition of multimodal spectral information.** The diagram enumerates modality subsets (single, pairwise, and all-modal combinations) and associates them with the dominant information type (unique vs. synergistic vs. redundant).

dundant $(R)$, unique $(U_m)$, and synergistic $(S)$ components:

$$\mathrm{I}(X;y) \approx R + \sum_{m=1}^{M} U_m + S. \qquad (5)$$

Guided by this formulation and multi-task routing paradigms (Ma et al., 2018), we partition the expert pool $\mathcal{C}$ into three functionally distinct subsets: $\mathcal{C} = \mathcal{C}_{\mathrm{sh}} \cup (\bigcup_m \mathcal{C}_m) \cup \mathcal{C}_{\mathrm{int}}$. Here, **Shared experts** ($\mathcal{C}_{\mathrm{sh}}$) capture redundancy $R$; **Modality-specific experts** ($\mathcal{C}_m$) extract unique constraints $U_m$; and crucially, **Interaction experts** ($\mathcal{C}_{\mathrm{int}}$) model synergy $S$, processing cross-modal dependencies that emerge only under joint consideration.

To operationalize this structure without intractable PID estimation, we impose lightweight regularizers that enforce the principle "redundant-consistent, unique-separable." We apply a consistency loss $\mathcal{L}_{\mathrm{sh}}$ to encourage shared experts to produce invariant representations across modalities, and a separability loss $\mathcal{L}_{\mathrm{sep}}$ to maintain the distinctiveness of modality-specific features:

$$\mathcal{L}_{\mathrm{struct}} = \mathbb{E}\Big[ \sum_{e \in \mathcal{C}_{\mathrm{sh}}} \mathrm{Dist}(o_i^{(e)}, o_j^{(e)}) \Big]$$
$$+ \mathbb{E}\Big[ \sum_{m=1}^{M} \sum_{e \in \mathcal{C}_m} \mathrm{Sep}(o_{i_m}^{(e)}, \{o^{(e)}\}_{m' \neq m}) \Big]. \qquad (6)$$

This formulation ensures that the router aligns token allocation with the physical nature of the evidence (as visualized in Figure 4), preventing parameter entanglement while preserving synergistic pathways. Further discussion of the PID-inspired expert partition and the corresponding structural regularizers is included in Section A.7 of the appendix.

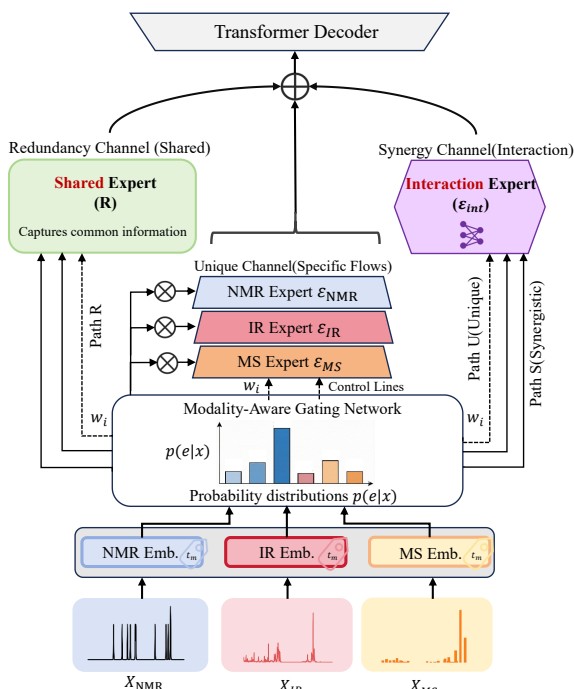

*Figure 4.* **Channelized modality-aware routing aligned with redundancy/unique/synergy pathways.** A modality-aware gating network consumes modality-conditioned representations (e.g., NMR/IR/MS embeddings) and produces routing distributions.

### 3.5. Heterogeneous Experts

**Heterogeneous Expert Spectrum.** To address the disparity in information density—where constraint-rich NMR tokens demand high nonlinearity while redundant IR tokens require only shallow processing—MM-Spectrum moves beyond homogeneous expert pools. We diversify the expert space into *Heavy* and *Light* experts, parameterized by varying capacities $\kappa_e$ (e.g., hidden dimensions). This design constructs a capacity spectrum, theoretically enabling the model to allocate computational resources proportional to the marginal utility of each token $\Delta_i$.

**Computation-Aware Regularization.** Standard routing does not spontaneously align expert capacity with token difficulty. To enforce a "Heavy-for-hard, Light-for-easy" specialization logic, we introduce a computation cost penalty. Let $c_e \propto \kappa_e$ be the cost assigned to expert $e$. The expected computational budget for a batch is minimized via:

$$\mathcal{L}_{\mathrm{cost}} = \frac{1}{B} \sum_{i=1}^{B} \sum_{e=1}^{E} p_{i,e}\, c_e. \qquad (7)$$

This soft constraint compels the router to activate Heavy experts only when the reduction in task loss $\mathcal{L}_{\mathrm{task}}$ outweighs the incurred cost $c_e$, effectively inducing a Pareto-efficient resource allocation. Figure 5 visually demonstrates this capacity–utility alignment, where expensive computa-

*Table 1.* **Modality-combination study.** Top-$K$ SMILES elucidation accuracy (%) under single-modality, bimodal, and full-modality.

| Modality Setting | Top-K Accuracy (%) | | | | | | | | | |
|---|---|---|---|---|---|---|---|---|---|---|
| | Top-1% | Top-2% | Top-3% | Top-4% | Top-5% | Top-6% | Top-7% | Top-8% | Top-9% | Top-10% |
| *Single-modality* | | | | | | | | | | |
| NMR (1H + 13C) | 69.83 ± 0.24 | 77.88 ± 0.16 | 80.86 ± 0.11 | 82.38 ± 0.38 | 83.35 ± 0.29 | 84.03 ± 0.34 | 84.52 ± 0.26 | 84.89 ± 0.32 | 85.08 ± 0.46 | 85.18 ± 0.12 |
| MS (all MS spectra) | 17.35 ± 0.30 | 23.82 ± 0.22 | 27.80 ± 0.41 | 30.08 ± 0.29 | 31.73 ± 0.15 | 32.90 ± 0.18 | 33.81 ± 0.40 | 34.52 ± 0.46 | 35.04 ± 0.17 | 35.31 ± 0.35 |
| IR (binned spectrum) | 23.22 ± 0.47 | 30.90 ± 0.42 | 35.01 ± 0.17 | 37.65 ± 0.14 | 39.44 ± 0.27 | 40.72 ± 0.21 | 41.73 ± 0.22 | 42.44 ± 0.29 | 42.88 ± 0.21 | 43.07 ± 0.19 |
| *Dual-modality* | | | | | | | | | | |
| NMR + MS (Baseline) | 59.15 ± 0.41 | 67.52 ± 0.46 | 70.75 ± 0.33 | 72.34 ± 0.20 | 73.34 ± 0.20 | 74.01 ± 0.33 | 74.58 ± 0.37 | 75.02 ± 0.25 | 75.35 ± 0.18 | 75.52 ± 0.25 |
| **NMR + MS (MM-Spectrum)** | **72.98 ± 0.18** | **83.02 ± 0.18** | **85.18 ± 0.20** | **86.14 ± 0.49** | **86.33 ± 0.29** | **86.60 ± 0.49** | **87.10 ± 0.18** | **87.24 ± 0.41** | **87.51 ± 0.38** | **87.66 ± 0.23** |
| NMR + IR (Baseline) | 64.88 ± 0.34 | 74.57 ± 0.32 | 78.76 ± 0.21 | 80.96 ± 0.12 | 82.40 ± 0.46 | 83.40 ± 0.16 | 84.09 ± 0.40 | 84.62 ± 0.11 | 84.96 ± 0.21 | 85.08 ± 0.19 |
| **NMR + IR (MM-Spectrum)** | **71.96 ± 0.18** | **81.50 ± 0.37** | **83.83 ± 0.48** | **84.91 ± 0.17** | **85.52 ± 0.48** | **85.99 ± 0.43** | **86.30 ± 0.12** | **86.55 ± 0.30** | **86.80 ± 0.20** | **86.99 ± 0.46** |
| MS + IR (Baseline) | 31.22 ± 0.11 | 39.68 ± 0.28 | 44.10 ± 0.18 | 46.77 ± 0.28 | 48.65 ± 0.32 | 50.06 ± 0.11 | 51.20 ± 0.12 | 52.10 ± 0.49 | 52.65 ± 0.30 | 52.93 ± 0.31 |
| **MS + IR (MM-Spectrum)** | **37.90 ± 0.34** | **47.08 ± 0.45** | **52.03 ± 0.15** | **54.90 ± 0.48** | **56.88 ± 0.25** | **58.48 ± 0.39** | **59.61 ± 0.14** | **60.43 ± 0.37** | **60.98 ± 0.21** | **61.26 ± 0.40** |
| *Tri-modality* | | | | | | | | | | |
| NMR + MS + IR (Baseline) | 44.29 ± 0.35 | 52.71 ± 0.28 | 56.17 ± 0.27 | 58.11 ± 0.24 | 59.39 ± 0.31 | 60.25 ± 0.48 | 60.97 ± 0.21 | 61.52 ± 0.31 | 61.82 ± 0.18 | 62.04 ± 0.47 |
| **NMR + MS + IR (MM-Spectrum)** | **76.04 ± 0.12** | **83.08 ± 0.11** | **85.65 ± 0.20** | **86.98 ± 0.13** | **87.83 ± 0.25** | **88.66 ± 0.24** | **89.31 ± 0.17** | **89.74 ± 0.40** | **90.01 ± 0.30** | **90.26 ± 0.13** |
| *Improvement* | 31.75 ± 0.39 | 30.37 ± 0.27 | 29.48 ± 0.37 | 28.87 ± 0.14 | 28.44 ± 0.41 | 28.41 ± 0.32 | 28.34 ± 0.31 | 28.22 ± 0.43 | 28.19 ± 0.29 | 28.22 ± 0.39 |

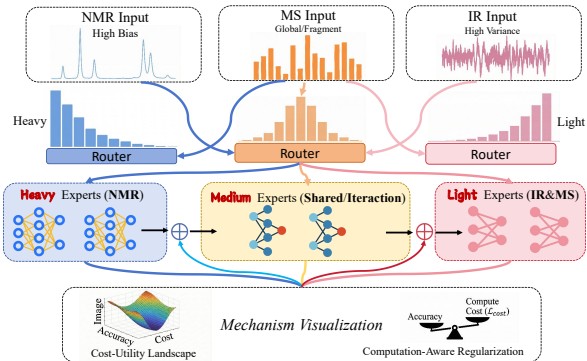

*Figure 5.* **Computation-aware regularization with heterogeneous experts: allocating expensive capacity to high-utility tokens.** MM-Spectrum instantiates an explicit cost–utility principle in conditional computation. This mechanism prevents long, low-density modalities from monopolizing compute, while preserving the ability to allocate high-capacity processing to decision-critical evidence, improving both stability and full-modality performance.

tion is reserved for high-utility tokens. The constrained-optimization interpretation of this objective and the resulting parameter and computation complexity are discussed in Sections A.8 and A.9 of the appendix.

**Curriculum-Scheduled Objective.** Standard MoE training faces a dilemma: strong balancing prevents collapse but hinders specialization, while weak balancing invites degeneracy (Zoph et al., 2022). We resolve this via a **Stability-to-Specialization Curriculum** (Bengio et al., 2009). The total objective is defined as:

$$\mathcal{L}_{\text{total}} = \mathcal{L}_{\text{task}} + \lambda_{\text{cost}}\mathcal{L}_{\text{cost}} + \lambda(t)\big(\mathcal{L}_{\text{bal}} + \mathcal{L}_{\text{ent}}\big), \quad (8)$$

where $\mathcal{L}_{\text{bal}}$ and $\mathcal{L}_{\text{ent}}$ correspond to standard load balancing (squared coefficient of variation) and entropy maximization regularizers (Shazeer et al., 2017; Fedus et al., 2022). Crucially, the regularization weight $\lambda(t)$ and router temperature $\tau(t)$ are annealed over training steps $t$. This schedule engen-

ders a dynamic optimization trajectory: initially, the training begins with a *coverage phase*, where high regularization enforces broad expert utilization to prevent early degenerate collapse; subsequently, as constraints decay, the process enters an *alignment phase* where the explicit modality bias (Sec. 3.3) guides tokens toward structurally appropriate subspaces; finally, the curriculum culminates in a *specialization phase* where $\mathcal{L}_{\text{task}}$ and $\mathcal{L}_{\text{cost}}$ dominate, driving the router toward precise, Pareto-efficient capacity allocation. Definitions of the balancing and entropy terms, together with the explicit schedules for $\lambda(t)$ and $\tau(t)$, are provided in Sections A.5 and A.6 of the appendix.

## 4. Experiments

### 4.1. Benchmark, Data, and Evaluation Protocol

**Benchmark and Modalities.** We evaluate MM-Spectrum on the multimodal spectroscopic benchmark introduced by Alberts et al. (2024), a standard testbed for *Molecular Structural Elucidation*. The dataset provides paired molecular structures and measurements across three modalities: **NMR** ($^1$H and $^{13}$C), **IR**, and **MS**. We adopt the canonical train/validation/test splits and preprocessing conventions to ensure comparability. Inputs are tokenized into modality-specific sequences and processed by an encoder–decoder architecture to predict SMILES strings. And all compared methods share the same backbone capacity and optimization budget, isolating the performance differences to the fusion and conditional-computation mechanisms. Supplementary cross-dataset evaluations are also conducted on the experimental SDBS database (Saito & Kinugasa, 2011) and the large-scale MMST dataset (Priessner et al., 2026) (see Appendix C).

**Metric.** We report Top-$K$ accuracy for SMILES-based elucidation (Weininger, 1988): for each input, the model

*Table 2.* **Missing-modality robustness.** Top-$K$ accuracy (%) under test-time missing-modality settings (S0–S6).

| Setting | Missing ratio | | | Baseline | | | MM-Spectrum | | | *Improvement* | | |
|---|---|---|---|---|---|---|---|---|---|---|---|---|
| | NMR | MS | IR | Top-1% | Top-5% | Top-10% | Top-1% | Top-5% | Top-10% | Top-1% | Top-5% | Top-10% |
| S0 | 0.0 | 0.0 | 0.0 | 44.29 ± 0.50 | 59.39 ± 0.17 | 62.04 ± 0.44 | 76.04 ± 0.18 | 87.83 ± 0.28 | 90.26 ± 0.16 | 31.75 ± 0.49 | 28.44 ± 0.34 | 28.22 ± 0.41 |
| S1 | 0.1 | 0.1 | 0.1 | 33.42 ± 0.45 | 44.79 ± 0.47 | 46.80 ± 0.49 | 65.01 ± 0.47 | 75.11 ± 0.37 | 76.43 ± 0.19 | 31.59 ± 0.30 | 30.33 ± 0.37 | 29.63 ± 0.17 |
| S2 | 0.3 | 0.3 | 0.3 | 17.29 ± 0.12 | 23.24 ± 0.10 | 24.31 ± 0.37 | 45.91 ± 0.26 | 54.21 ± 0.17 | 55.40 ± 0.19 | 28.62 ± 0.33 | 30.97 ± 0.16 | 31.09 ± 0.44 |
| S3 | 0.6 | 0.6 | 0.6 | 4.28 ± 0.14 | 5.77 ± 0.24 | 6.05 ± 0.48 | 21.38 ± 0.41 | 26.20 ± 0.42 | 27.04 ± 0.46 | 17.10 ± 0.32 | 20.42 ± 0.47 | 21.00 ± 0.34 |
| S4 | 0.6 | 0.1 | 0.1 | 14.88 ± 0.46 | 20.02 ± 0.17 | 20.94 ± 0.17 | 28.96 ± 0.47 | 33.47 ± 0.21 | 34.02 ± 0.29 | 14.09 ± 0.40 | 13.44 ± 0.47 | 13.09 ± 0.28 |
| S5 | 0.1 | 0.6 | 0.1 | 14.89 ± 0.28 | 19.97 ± 0.34 | 20.89 ± 0.35 | 62.16 ± 0.25 | 72.68 ± 0.49 | 74.11 ± 0.29 | 47.28 ± 0.18 | 52.71 ± 0.17 | 53.22 ± 0.12 |
| S6 | 0.1 | 0.1 | 0.6 | 21.37 ± 0.48 | 28.82 ± 0.37 | 30.10 ± 0.11 | 54.56 ± 0.35 | 65.84 ± 0.20 | 67.63 ± 0.50 | 33.19 ± 0.38 | 37.03 ± 0.18 | 37.53 ± 0.49 |

produces $K$ candidate SMILES and a prediction is counted as correct if the ground-truth appears in the top $K$ list. Top-1 measures ranking sharpness and single-shot correctness, while larger $K$ reflects candidate coverage and recoverability under downstream filtering or re-ranking.

**Compared methods.** We compare: **Dense concatenation** (the standard baseline employing naive concatenation of multimodal token sequences under a standard Transformer) (Alberts et al., 2024), and **MM-Spectrum** (our modality-aware sparse MoE encoder with structured expert subspaces, heterogeneous capacities, and computation-cost regularization). For a fair comparison, the decoder and non-MoE components are kept identical across methods.

**Implementation Details.** We implement all models using PyTorch on **NVIDIA A40 (40GB) GPUs**. The backbone is a standard Transformer Encoder-Decoder with $L = 6$ layers, $H = 8$ attention heads, and hidden dimension $d_{model} = 512$. For the **MM-Spectrum** encoder, we configure the MoE layers with a total of $E = 8$ experts (partitioned into shared, specific, and interaction groups as described) and employ Top-$k = 2$ gating with a capacity factor of $C = 1.25$ to balance load. During inference, we use beam search with a beam width of 10. Complete architectural hyperparameters, routing configurations, preprocessing settings, and decoding details are reported in Sections B.2, B.4 and C.1 of the appendix.

### 4.2. Modality Combination Study

To investigate whether MM-Spectrum mitigates negative transfer induced by spectral heterogeneity, we evaluate elucidation accuracy across all seven combinations of NMR, IR, and MS (unimodal, bimodal, and full-modality). We compare our method against a standard **Dense** baseline that uses early concatenation, keeping backbone capacities identical. As shown in Table 1, the single-modality baseline reveals significant imbalance, with NMR providing the strongest standalone constraints (69.83%). Crucially, the Dense concatenation baseline exhibits **catastrophic negative transfer** in the full-modality regime: instead of benefiting from additional evidence, its performance collapses to 44.29%,

significantly underperforming the unimodal NMR baseline. This confirms our hypothesis that naive fusion allows long, low-density modalities (IR/MS) to overwhelm optimization and dilute high-fidelity signals.

In contrast, **MM-Spectrum** yields consistent Pareto improvements across all settings. Most notably, it successfully reverses the full-modality collapse (44.29% → 76.04%), transforming the additional noisy modalities from a source of interference into a source of synergy. The substantial gains in Top-1 accuracy indicate that our structured expert allocation effectively disentangles conflicting gradients, allowing the model to sharpen ranking precision by selectively integrating complementary evidence rather than succumbing to attention dilution. Additional comparisons against cross-attention, contrastive-alignment, and gradient-modulation baselines are provided in Section C.2 of the appendix. And the large-scale evaluation on MMST is reported in Section C.3.

### 4.3. Missing-Modality Robustness

To evaluate model reliability under realistic instrumentation failures, we test elucidation performance across seven inference-time missingness settings (S0–S6), ranging from balanced dropout to heavy single-modality loss, without any test-time adaptation. As detailed in Table 2, the baseline exhibits brittle behavior with sharp performance drops as missingness increases, suggesting implicit co-adaptation that fails when specific inputs are absent. In contrast, **MM-Spectrum** demonstrates smoother degradation and superior candidate recoverability (Top-10). This resilience indicates that our modality-aware routing successfully establishes reconfigurable inference pathways, capable of dynamically re-allocating computational budget to the remaining reliable evidence rather than collapsing under distribution shift.

### 4.4. Generalization to Experimental Spectra

To evaluate the robustness and practical applicability of **MM-Spectrum** on empirical laboratory measurements, we conduct extensive evaluations on the Spectral Database for Organic Compounds (SDBS) (Saito & Kinugasa, 2011). We

*Table 3.* **Stratified evaluation by molecular complexity.** Top-$K$ accuracy (%) across Heavy Atom Count (HAC) bins. MM-Spectrum yields larger relative improvements on more complex molecules.

| Bucket | HAC range | #Samples | Baseline | | | MM-Spectrum | | | *Improvement* | | |
|---|---|---|---|---|---|---|---|---|---|---|---|
| | | | Top-1% | Top-5% | Top-10% | Top-1% | Top-5% | Top-10% | Top-1% | Top-5% | Top-10% |
| HAC_S (Small) | $\leq 15$ | 14821 | $63.46_{\pm 0.22}$ | $78.48_{\pm 0.37}$ | $81.44_{\pm 0.28}$ | $82.28_{\pm 0.42}$ | $91.30_{\pm 0.35}$ | $92.45_{\pm 0.32}$ | $18.82_{\pm 0.30}$ | $12.82_{\pm 0.38}$ | $11.01_{\pm 0.12}$ |
| HAC_M (Medium) | 16–25 | 35985 | $46.89_{\pm 0.29}$ | $63.16_{\pm 0.16}$ | $66.04_{\pm 0.39}$ | $77.52_{\pm 0.32}$ | $88.03_{\pm 0.37}$ | $89.19_{\pm 0.44}$ | $30.63_{\pm 0.34}$ | $24.87_{\pm 0.25}$ | $23.15_{\pm 0.32}$ |
| HAC_L (Large) | $\geq 26$ | 28635 | $29.93_{\pm 0.35}$ | $43.56_{\pm 0.35}$ | $46.48_{\pm 0.45}$ | $69.44_{\pm 0.29}$ | $81.50_{\pm 0.23}$ | $82.86_{\pm 0.38}$ | $39.51_{\pm 0.31}$ | $37.94_{\pm 0.33}$ | $36.38_{\pm 0.16}$ |

*Table 4.* **Generalization to real-world experimental spectra.** Top-$K$ SMILES elucidation accuracy (%) on the SDBS database under from-scratch and pre-training & fine-tuning (FT) protocols.

| Protocol | Setting | Top-K Accuracy (%) | | |
|---|---|---|---|---|
| | | Top-1% | Top-5% | Top-10% |
| *Scratch* | NMR | $18.15_{\pm 0.31}$ | $31.74_{\pm 0.25}$ | $33.69_{\pm 0.42}$ |
| | MS | $5.38_{\pm 0.18}$ | $11.07_{\pm 0.22}$ | $13.92_{\pm 0.15}$ |
| | IR | $13.21_{\pm 0.29}$ | $25.48_{\pm 0.34}$ | $26.61_{\pm 0.27}$ |
| | Baseline (Full) | $14.10_{\pm 0.33}$ | $27.13_{\pm 0.41}$ | $31.53_{\pm 0.38}$ |
| | **Ours (Full)** | $\mathbf{26.67}_{\pm 0.24}$ | $\mathbf{38.91}_{\pm 0.19}$ | $\mathbf{47.37}_{\pm 0.26}$ |
| *Pretrain & FT* | Baseline (Full) | $37.56_{\pm 0.36}$ | $57.15_{\pm 0.28}$ | $59.84_{\pm 0.31}$ |
| | **Ours (Full)** | $\mathbf{59.52}_{\pm 0.21}$ | $\mathbf{73.95}_{\pm 0.25}$ | $\mathbf{76.32}_{\pm 0.19}$ |

benchmark our model under two distinct paradigms: (1) training *from scratch* directly on the experimental dataset, and (2) utilizing a *pre-training and fine-tuning* protocol, where the network is initialized on simulated spectra and subsequently adapted to real laboratory observations.

As summarized in Table 4, the single-modality baselines trained from scratch reveal the inherent complexity of uncurated experimental signals, with NMR remaining the most constraint-rich channel. Crucially, when combining all channels under full-modality inputs, the naive dense concatenation baseline exhibits a notable performance degradation compared to the standalone NMR setup, dropping from 18.15% to 14.10% Top-1 accuracy. In contrast, **MM-Spectrum** successfully circumvents this multi-source interference. When training from scratch, it completely reverses the full-modality degradation, boosting Top-1 accuracy to 26.67% and outperforming the dense multi-spectral baseline. Furthermore, under the pre-training and fine-tuning protocol, **MM-Spectrum** capitalizes on the large-scale prior representations to achieve a substantial performance leap.

### 4.5. Stratified Evaluation by Molecular Complexity

To probe performance beyond aggregate metrics, we stratify test molecules by Heavy Atom Count (HAC), a proxy for structural complexity and search-space ambiguity. We partition the data into small ($HAC_S$), medium ($HAC_M$), and large ($HAC_L$) bins based on ground-truth structures. Results in Table 3 reveal a clear complexity-dependent degradation for the Dense concatenation baseline. In contrast, **MM-Spectrum** achieves consistent improvements across all strata, with the most substantial relative gains observed

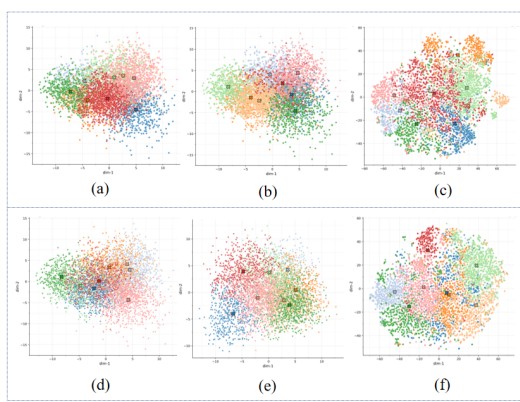

*Figure 6.* **Representation evolution over training.** Dense concatenation (top row) vs. MM-Spectrum (bottom row) at multiple training steps. MM-Spectrum exhibits more coherent and stable structure in the learned representation space, consistent with reduced cross-modal interference.

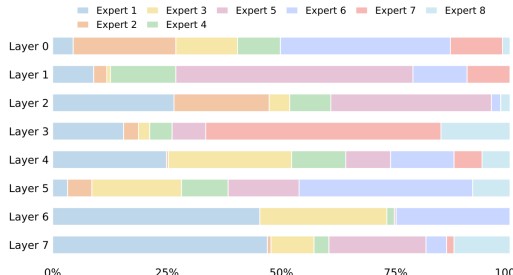

*Figure 7.* **Layer-wise expert utilization.** 100% stacked bar charts of Top-1 expert fractions per layer. Different layers display distinct utilization profiles, indicating layer-dependent specialization rather than trivial collapse or uniform routing.

on $HAC_M$ and $HAC_L$. This trend corroborates the efficacy of our capacity–utility principle: difficult instances trigger high-capacity and interaction-specific experts, effectively scaling computational resources to resolve greater posterior uncertainty. Complementary sensitivity analyses of expert capacity and routing sparsity are reported in Section C.5 of the appendix.

### 4.6. Interpretability and Training Dynamics

**Representation Evolution.** Visualizing encoder representations (Figure 6) reveals that while the Dense concatenation baseline exhibits progressive mixing and instability—indicative of entanglement—MM-Spectrum develops

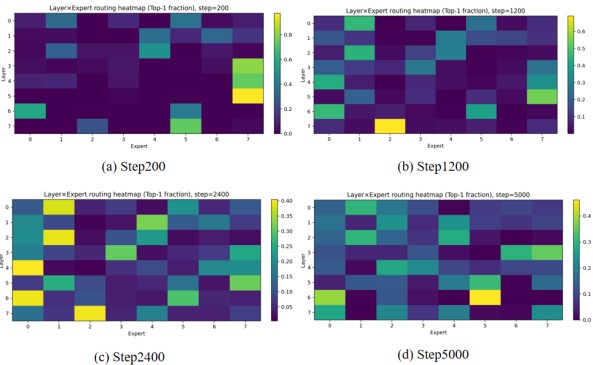

**Figure 8.** **Routing dynamics (Layer×Expert heatmaps).** Top-1 expert occupancy across layers at multiple training steps. The routing pattern transitions from early exploration to structured specialization and stabilizes at later stages, providing direct evidence of emergent expert division of labor.

a stable, well-separated organization. This confirms that modality-aware routing and structured subspaces effectively reduce interference and support disentangled evidence aggregation.

**Routing Dynamics and Specialization.** We further analyze expert behavior via occupancy heatmaps (Figure 8) and layer-wise profiles (Figure 7). Routing trajectories evolve from diffuse early exploration to stable, persistent hotspots, validating our curriculum-driven transition from coverage to specialization. Crucially, expert utilization varies distinctively across depths without degenerating into collapse or uniformity, implying that different layers autonomously acquire distinct functional roles (feature alignment vs. decision transformation) driven by the capacity–utility principle.

### 4.7. Ablation Studies

*Table 5.* **Ablation: routing signals.** Impact of Router-aware signals and Tag routing on Top-$K$ accuracy (%).

| Router-aware | Tag-routing | Top-K Accuracy (%) | | |
|:---:|:---:|:---:|:---:|:---:|
| | | **Top-1%** | **Top-5%** | **Top-10%** |
| ✓ | ✓ | **76.04** ± **0.20** | **87.83** ± **0.39** | **90.26** ± **0.13** |
| ✗ | ✓ | 72.57 ± 0.26 | 86.39 ± 0.48 | 88.93 ± 0.39 |
| ✓ | ✗ | 71.23 ± 0.25 | 86.06 ± 0.16 | 88.83 ± 0.15 |
| ✗ | ✗ | 70.11 ± 0.30 | 84.43 ± 0.31 | 86.06 ± 0.34 |

**Routing signals: Router-aware and Tag routing.** We first assess how making modality cues explicit affects specialization. Table 5 shows that removing either Router-aware signals or Tag routing consistently reduces Top-$K$, with the largest relative degradation at Top-1. This pattern indicates that explicit routing supervision improves not only candidate coverage but also ranking precision, consistent with faster and cleaner formation of expert responsibilities.

*Table 6.* **Ablation: expert subspaces and dynamic activation.** Effects of Shared experts, Interaction experts, and Dynamic activation on Top-$K$ accuracy (%).

| Shared | Interaction | Dynamic | Top-K Accuracy (%) | | |
|:---:|:---:|:---:|:---:|:---:|:---:|
| | | | **Top-1%** | **Top-5%** | **Top-10%** |
| ✓ | ✓ | ✓ | **76.04** ± **0.15** | **87.83** ± **0.32** | **90.26** ± **0.21** |
| ✓ | ✓ | ✗ | 74.41 ± 0.28 | 86.54 ± 0.41 | 88.83 ± 0.19 |
| ✓ | ✗ | ✓ | 74.39 ± 0.35 | 86.34 ± 0.22 | 88.63 ± 0.45 |
| ✗ | ✓ | ✓ | 75.01 ± 0.12 | 86.90 ± 0.33 | 89.20 ± 0.27 |
| ✗ | ✗ | ✓ | 72.70 ± 0.40 | 84.65 ± 0.18 | 87.05 ± 0.36 |

**Expert Subspaces and Dynamic Activation.** Decomposing the expert space (Table 6) demonstrates that *Interaction* experts are pivotal for capturing cross-modal synergy, validating the need for dedicated parameter pathways over naive fusion. *Shared* experts concurrently facilitate redundant evidence transfer, while Dynamic activation further boosts robustness, confirming the benefit of adaptive computation allocation under sample-level variability.

*Table 7.* **Ablation: heterogeneous experts and regularization.** Effects of heterogeneous capacities (Hetero) and computation-aware regularization on Top-$K$ accuracy (%).

| Hetero. | Regularizer | Top-K Accuracy (%) | | |
|:---:|:---:|:---:|:---:|:---:|
| | | **Top-1%** | **Top-5%** | **Top-10%** |
| ✓ | ✓ | **76.04** ± **0.14** | **87.83** ± **0.29** | **90.26** ± **0.18** |
| ✓ | ✗ | 74.81 ± 0.35 | 86.61 ± 0.22 | 89.21 ± 0.41 |
| ✗ | ✓ | 74.07 ± 0.19 | 85.93 ± 0.38 | 88.65 ± 0.26 |
| ✗ | ✗ | 74.24 ± 0.44 | 86.08 ± 0.15 | 88.65 ± 0.32 |

**Heterogeneous Capacities and Regularization.** Finally, Table 7 substantiates the synergy between heterogeneous expert capacities and computation-aware regularization. The combined improvement indicates that meaningful capacity diversity enables the regularizer to effectively match computational budget to token complexity. These findings strongly corroborate our proposed *capacity–utility* principle for conditional computation in multispectral elucidation.

## 5. Conclusion

We presented **MM-Spectrum**, a sparse MoE framework tailored to resolve the optimization instability in multimodal *Molecular Structural Elucidation* arising from severe spectral heterogeneity. By synergizing spectroscopy-aware compression, explicit modality routing, and structured expert subspaces with curriculum-driven heterogeneous computation, MM-Spectrum enforces Pareto-efficient specialization. This design mitigates imbalance-induced conflicts and successfully resolves full-modality collapse. While not intended to outcompete highly specialized single-modality models, MM-Spectrum yields substantial gains in the full-modality regime, establishing a principled paradigm for extracting faithful cross-spectral synergy.

## Acknowledgements

We sincerely thank the anonymous reviewers for their insightful comments and constructive suggestions. This research is supported by the National Natural Science Foundation of China Project (No. 623B2086), Sponsored by CCF-GHFund (No. OF 2026005), Sponsored by CIPS-SMP-Zhipu Large Model Fund, Ant Group, and TeleAI of China Telecom.

## Impact Statement

This paper presents work whose goal is to advance the field of multimodal machine learning for scientific discovery, specifically in chemical structure elucidation. Potential societal consequences include accelerating drug discovery and materials science research. We do not foresee any negative ethical impacts or adverse societal consequences from this work.

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

# Appendix: MM-Spectrum: Multispectral Molecular Structural Elucidation under Modality Imbalance with a Stable MoE Framework

## A. Theoretical Background and Derivations

### A.1. Notation and Task Objective

We consider multimodal spectra-to-structure generation with $M$ modalities. For each modality $m \in \{1, \ldots, M\}$, the tokenized spectral sequence is $x^{(m)} = (x_1^{(m)}, \ldots, x_{L_m}^{(m)})$ with length $L_m$. Let $X = \{x^{(m)}\}_{m=1}^M$ denote the multimodal input and $y = (y_1, \ldots, y_T) \in \mathcal{Y}$ the target SMILES sequence. An encoder–decoder model parameterized by $\theta$ defines

$$p_\theta(y \mid X) = \prod_{t=1}^T p_\theta(y_t \mid y_{<t}, X), \qquad \mathcal{L}_{\text{task}}(\theta) = -\mathbb{E}_{(X,y)\sim\mathcal{D}}\Big[\sum_{t=1}^T \log p_\theta(y_t \mid y_{<t}, X)\Big]. \tag{9}$$

**(a) Existing Methods: Naive Concatenation**

**(b) Our Method: MM-Spectrum with MoE-Encoder**

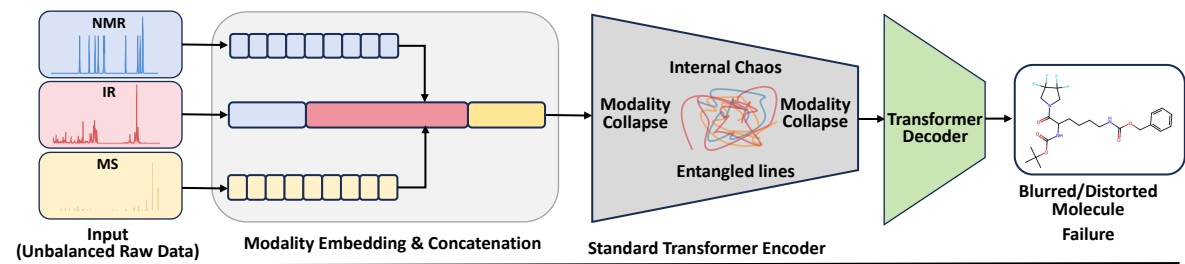

*Figure 9.* **Visualizing the mechanism of failure versus the mechanism of synergy. (a) The Pathology of Naive Concatenation:** Treating heterogeneous signals (NMR/IR/MS) as a uniform sequence leads to *Modality Collapse*. The shared parameter space succumbs to gradient conflicts (visualized as "Internal Chaos"), where high-redundancy modalities overwhelm high-density constraints, resulting in entangled representations and distorted molecular predictions. **(b) The Solution via MM-Spectrum:** We resolve this by introducing a **Structured MoE Encoder**. Through a modality-aware gating mechanism $G(x)$, the model dynamically disentangles information flows by routing tokens to functionally specialized modules—allocating *Heavy Experts* for complex topology inference and *Light Experts* for noise filtering—thereby restoring optimization stability and generation precision.

In this appendix we formalize: (i) why naive early-fusion concatenation can *degrade* in the full-modality regime, (ii) how modality-aware routing stabilizes specialization, and (iii) how structured experts and cost-aware regularization yield a Pareto-efficient allocation under multispectral imbalance.

### A.2. A Multi-Objective View of Multispectral Training

A useful lens is to treat multispectral learning as optimizing a sum of modality-induced objectives. Let $C_m(\theta)$ be the expected loss contribution "attributable" to modality $m$ (e.g., the loss induced when modality $m$ is present and the others are treated as nuisance factors). A generic multi-objective surrogate can be written as

$$C(\theta) = \sum_{m=1}^M w_m\, C_m(\theta), \qquad w_m \geq 0, \ \sum_m w_m = 1. \tag{10}$$

Let $g_m = \nabla_\theta C_m(\theta)$. Gradient *compatibility* between modalities $i$ and $j$ can be measured by cosine similarity

$$\cos(g_i, g_j) = \frac{\langle g_i, g_j \rangle}{\|g_i\| \|g_j\|}. \tag{11}$$

Negative transfer occurs when cross-modal updates conflict, i.e., $\cos(g_i, g_j) < 0$, so that improving one modality tends to degrade the other. In multispectral spectroscopy, conflicts are exacerbated by *imbalance*: modalities can differ by orders of magnitude in length ($L_m$), noise, and per-token semantic density.

### A.3. Why Naive Concatenation Can Collapse Under Multispectral Imbalance

The standard baseline forms a single early-fusion sequence

$$x^{(\text{cat})} = \text{Concat}\big(x^{(1)}, \dots, x^{(M)}\big), \qquad L = \sum_{m=1}^{M} L_m. \tag{12}$$

This design implicitly allocates attention budget and gradient updates roughly in proportion to token counts. When $L_m$ is severely imbalanced, "long but weak" modalities can dominate the training signal even if their *marginal utility per token* is low.

**A density-weighted intuition.** Let $\rho_m$ be an abstract (unobserved) measure of per-token information density for modality $m$ with respect to the task. Under early fusion, the *effective* contribution of modality $m$ to the overall update is influenced by both its token count and density. A simplified heuristic is that the expected magnitude of gradients contributed by modality $m$ scales with the number of tokens that backpropagate through shared parameters:

$$\mathbb{E}\big[\|g_m\|\big] \propto L_m \cdot \bar{\delta}_m, \tag{13}$$

where $\bar{\delta}_m$ summarizes average per-token backprop signal strength. In multispectral spectroscopy, it is common that $L_{\text{IR}} \gg L_{\text{NMR}}$ but $\bar{\delta}_{\text{IR}} < \bar{\delta}_{\text{NMR}}$, so $L_m$ can dominate even when $\rho_m$ is low. This yields attention dilution (high-density constraints become "background") and can also amplify cross-modal gradient conflicts, pushing the optimizer toward a suboptimal compromise solution.

**Key implication.** Full-modality improvement is not guaranteed by simply adding modalities. Instead, stable synergy requires an explicit mechanism that can (i) prevent low-density tokens from monopolizing updates, and (ii) structurally separate conflicting gradient components so that specialization can emerge.

### A.4. Modality-Aware Routing as an Explicit "Identity + Content" Decomposition

**Standard MoE routing.** A sparse MoE layer replaces the dense FFN with experts $\{\text{FFN}_e\}_{e=1}^{E}$ and a router that produces gating probabilities $p_{i,e}$ for each token representation $h_i \in \mathbb{R}^d$:

$$p_{i,e} = \text{softmax}\Big(\frac{w_e^\top h_i}{\tau}\Big)_e, \qquad o_i = \sum_{e \in \text{TopK}(p_{i,:,k})} p_{i,e} \, \text{FFN}_e(h_i). \tag{14}$$

In heterogeneous spectroscopy, content-only routing forces the router to *infer* modality identity from token statistics, which is unnecessary and unstable when modalities have drastically different distributions.

**MM-Spectrum: additive modality injection.** For each token $i$, let $m(i)$ be its modality index. We inject explicit modality information via a learnable tag embedding $e_{m(i)}$ and a modality bias $b_{m(i)}$:

$$\tilde{h}_i = h_i + e_{m(i)} + b_{m(i)}. \tag{15}$$

The router then gates on $\tilde{h}_i$:

$$p_{i,e} = \text{softmax}\Big(\frac{w_e^\top \tilde{h}_i}{\tau}\Big)_e, \qquad o_i = \sum_{e \in \text{TopK}(p_{i,:,k})} p_{i,e} \, \text{FFN}_e(h_i). \tag{16}$$

**Interpretation: logit bias and stabilized coarse separation.** Define logits $z_{i,e} = w_e^\top \tilde{h}_i$. Then

$$z_{i,e} = w_e^\top h_i \; + \; w_e^\top (e_{m(i)} + b_{m(i)}). \tag{17}$$

The second term is a modality-dependent expert preference that is *disentangled* from content. Geometrically, Eq. (15) induces a translation in the router input space, so tokens from different modalities become separable even when their content features overlap early in training. This yields a stable "coverage" behavior in early optimization (reducing collapse risk), while still allowing fine-grained content-based routing within each modality.

## A.5. Load Balancing and Entropy Regularization

Sparse MoE typically uses auxiliary losses to avoid expert collapse. Let $\mathcal{K}_i = \text{TopK}(p_{i,:}, k)$ be the activated set for token $i$ in a minibatch of $B$ tokens. Define expert *importance* and *load*:

$$\text{Imp}_e = \sum_{i=1}^{B} p_{i,e}, \qquad \text{Load}_e = \sum_{i=1}^{B} \mathbf{1}[e \in \mathcal{K}_i]. \tag{18}$$

We use the squared coefficient of variation ($\text{CV}^2$) to penalize concentration:

$$\mathcal{L}_{\text{bal}} = \text{CV}^2(\text{Imp}) + \text{CV}^2(\text{Load}) = \frac{\text{Var}(\text{Imp})}{(\mathbb{E}[\text{Imp}])^2} + \frac{\text{Var}(\text{Load})}{(\mathbb{E}[\text{Load}])^2}. \tag{19}$$

Additionally, to encourage exploration early in training, we regularize router entropy:

$$\mathcal{L}_{\text{ent}} = -\frac{1}{B} \sum_{i=1}^{B} \sum_{e=1}^{E} p_{i,e} \log p_{i,e}. \tag{20}$$

## A.6. Stability-to-Specialization Curriculum: Scheduling $\lambda(t)$ and $\tau(t)$

In multispectral settings, strong balancing is beneficial early (preventing collapse) but can become harmful late (preventing specialization and synergy). We therefore schedule the strength of auxiliary regularizers and the router temperature over training steps $t$:

$$\mathcal{L}_{\text{total}} = \mathcal{L}_{\text{task}} + \lambda_{\text{cost}} \mathcal{L}_{\text{cost}} + \lambda(t)\big(\mathcal{L}_{\text{bal}} + \mathcal{L}_{\text{ent}}\big), \tag{21}$$

where $\lambda(t)$ is annealed from a large value to a small value. We use a smooth cosine decay (other monotone decays are valid):

$$\lambda(t) = \lambda_0 \cdot \frac{1}{2}\Big(1 + \cos(\pi \cdot \min(1, t/T_\lambda))\Big). \tag{22}$$

Similarly, we anneal temperature to move from exploration to confident routing:

$$\tau(t) = \max\Big(\tau_{\min}, \; \tau_0 \cdot \exp(-t/T_\tau)\Big). \tag{23}$$

**Three-phase dynamics.** This induces an interpretable trajectory: (i) **coverage phase**: large $\lambda(t)$ and higher $\tau(t)$ encourage broad usage; (ii) **alignment phase**: decreasing $\lambda(t)$ allows modality bias to form stable channels; (iii) **specialization phase**: $\mathcal{L}_{\text{task}}$ and $\mathcal{L}_{\text{cost}}$ dominate, yielding sharp and cost-effective routing.

## A.7. Structured Expert Space via PID-Induced Inductive Bias

We adopt Partial Information Decomposition (PID) as an inductive bias to structure expert roles. Let $R$ denote redundant information shared across modalities, $U_m$ unique information of modality $m$, and $S$ synergistic information:

$$I(X; y) \approx R + \sum_{m=1}^{M} U_m + S. \tag{24}$$

Guided by Eq. (24), we partition experts into three families:

$$\mathcal{C} = \mathcal{C}_{\text{sh}} \cup \Big(\bigcup_{m=1}^{M} \mathcal{C}_m\Big) \cup \mathcal{C}_{\text{int}}, \tag{25}$$

where $\mathcal{C}_{\text{sh}}$ are **shared** experts (redundancy), $\mathcal{C}_m$ are **modality-specific** experts (unique constraints), and $\mathcal{C}_{\text{int}}$ are **interaction** experts (synergy).

**Lightweight structural regularizers.** We impose weak, implementation-friendly constraints that encourage "redundant-consistent, unique-separable" behavior without explicitly estimating PID:

$$\mathcal{L}_{\text{struct}} = \mathcal{L}_{\text{sh}} + \mathcal{L}_{\text{sep}}. \tag{26}$$

For shared experts, we encourage cross-modality consistency for representations associated with the same molecule:

$$\mathcal{L}_{\text{sh}} = \mathbb{E}\Big[ \sum_{e \in \mathcal{C}_{\text{sh}}} \big\| \mu_{\text{NMR}}^{(e)} - \mu_{\text{IR}}^{(e)} \big\|_2^2 + \big\| \mu_{\text{NMR}}^{(e)} - \mu_{\text{MS}}^{(e)} \big\|_2^2 \Big], \tag{27}$$

where $\mu_m^{(e)}$ denotes the mean pooled expert-$e$ output for modality $m$ (within a sample or minibatch). For modality-specific experts, we enforce separability by a margin objective:

$$\mathcal{L}_{\text{sep}} = \mathbb{E}\Big[ \sum_{m=1}^{M} \sum_{e \in \mathcal{C}_m} \max\big(0,\ \gamma - d(\mu_m^{(e)}, \mu_{\neg m}^{(e)})\big) \Big], \tag{28}$$

where $\mu_{\neg m}^{(e)}$ is the mean pooled output over other modalities and $d(\cdot, \cdot)$ can be cosine distance or $\ell_2$ distance. In practice, $\mathcal{L}_{\text{struct}}$ is assigned a small weight so it shapes the geometry without overriding the main task loss.

## A.8. Heterogeneous Experts and Computation-Aware Regularization

**Heterogeneous capacity spectrum.** Different spectral tokens have different marginal utilities. We instantiate a spectrum of expert capacities via a per-expert descriptor $\kappa_e$ (e.g., hidden width in FFN), allowing "heavy" and "light" experts:

$$\text{FFN}_e(h) = W_2^{(e)} \sigma\big(W_1^{(e)} h\big), \qquad \kappa_e := \dim(W_1^{(e)} h). \tag{29}$$

**Cost-aware routing as a soft constrained optimization.** Associate each expert with a cost $c_e \propto \kappa_e$. We penalize the *expected* routing mass sent to expensive experts:

$$\mathcal{L}_{\text{cost}} = \frac{1}{B} \sum_{i=1}^{B} \sum_{e=1}^{E} p_{i,e}\, c_e. \tag{30}$$

Then minimizing $\mathcal{L}_{\text{task}} + \lambda_{\text{cost}} \mathcal{L}_{\text{cost}}$ is a Lagrangian relaxation of a constrained problem:

$$\min_{\theta} \ \mathcal{L}_{\text{task}}(\theta) \quad \text{s.t.} \quad \mathbb{E}\Big[ \sum_e p_{i,e} c_e \Big] \leq \mathcal{B}, \tag{31}$$

which encourages a Pareto-efficient allocation: high-utility tokens are allowed to use heavy experts, while low-utility redundant tokens are discouraged from monopolizing expensive capacity.

## A.9. Compute and Parameter Complexity

We provide a simple comparison between a dense Transformer FFN and a Top-$k$ MoE FFN. Let the dense FFN cost per token be $C_{\text{ffn}}$ (FLOPs). In MoE, only $k$ experts are executed, so ignoring dispatch overhead:

$$C_{\text{moe}} \approx k \cdot \bar{C}_{\text{expert}} + C_{\text{router}}, \tag{32}$$

where $\bar{C}_{\text{expert}}$ is the average cost of an expert and $C_{\text{router}}$ is small. If experts are heterogeneous, $\bar{C}_{\text{expert}}$ becomes routing-dependent; in that case $\mathcal{L}_{\text{cost}}$ (Eq. (30)) directly regularizes the expected compute.

Parameter-wise, MoE increases capacity approximately linearly with $E$:

$$P_{\text{moe}} \approx P_{\text{shared}} + E \cdot P_{\text{expert}}, \tag{33}$$

while keeping per-token compute controlled by $k$ (not $E$).

## B. Additional Implementation Details and Reproducibility

### B.1. Tokenization and Compression Operators

To align sequence statistics across modalities, we apply modality-specific preprocessing $\phi_m$:

$$\tilde{x}^{(m)} = \phi_m(x^{(m)}), \qquad \tilde{X} = \{\tilde{x}^{(m)}\}_{m=1}^M. \tag{34}$$

For high-density NMR, we preserve discrete constraints ($\phi_{\text{NMR}} \approx \text{Id}$). For high-redundancy IR/MS, we use local binning (IR) and peak filtering (MS) to suppress background tokens. Modality tags are appended (or added as embeddings) so each token retains explicit modality identity.

*Table 8.* **Modality preprocessing and sequence configuration (illustrative).** Operators are chosen to reduce redundancy while preserving chemically critical evidence.

| Modality | Token type | Compression $\phi_m$ | Max length |
|---|---|---|---|
| NMR ($^1$H/$^{13}$C) | symbolic / constraint-rich | identity / conservative parsing | $L_{\max}^{\text{NMR}}$ |
| IR | dense / autocorrelated | local binning / aggregation | $L_{\max}^{\text{IR}}$ |
| MS | sparse peaks + noise tail | top-$k$ peak filtering | $L_{\max}^{\text{MS}}$ |

### B.2. MoE Configuration and Routing Schedules

We use Top-$k$ gating and a capacity factor to reduce dispatch overflow. The router temperature $\tau(t)$ and regularizer strength $\lambda(t)$ follow Eqs. (22)–(23). All ablations keep the decoder and non-MoE components fixed.

*Table 9.* **Key MoE hyperparameters.**

| Hyperparameter | Value |
|---|---|
| #experts $E$ | 8 |
| Top-$k$ | 2 |
| Capacity factor $C$ | 1.25 |
| Aux losses | $\mathcal{L}_{\text{bal}}, \mathcal{L}_{\text{ent}}, \mathcal{L}_{\text{cost}}$ |
| Schedules | $\lambda(t)$ (cosine), $\tau(t)$ (exp or cosine) |
| Expert families | shared / modality-specific / interaction |

### B.3. Missing-Modality Protocol

For missing-modality evaluation, we fix a trained model and only alter test-time inputs. Given missing ratios $(r_{\text{NMR}}, r_{\text{MS}}, r_{\text{IR}})$, we randomly drop (mask) the corresponding fraction of each modality's tokens (or remove the modality channel), while preserving modality tags for remaining tokens. This evaluates whether routing can re-allocate compute to reliable evidence under distribution shift, without any test-time adaptation.

### B.4. Evaluation: Top-$K$ Accuracy with Beam Search

We decode with beam search (beam width $B$) and report Top-$K$ accuracy: a prediction is correct if the ground-truth SMILES appears among the top $K$ candidates. Top-1 reflects single-shot ranking accuracy, whereas larger $K$ values measure whether the correct structure remains recoverable within a candidate set.

### B.5. Data Preprocessing Details

For the dataset from Alberts et al. (2024), we applied the following modality-specific preprocessing:

- **NMR:** Discretized into symbolic tokens representing chemical shift ranges (0.1 ppm bins) and multiplicities. Max length truncated to 256.

- **IR:** Binned into 1024 distinct frequency intervals. Intensities are normalized to $[0, 1]$.

- **MS:** Top-100 peak filtering is applied based on relative intensity.

## C. Detailed Experimental Setup and Results

### C.1. Hyperparameter Configuration

To facilitate full reproducibility of our results, we provide the comprehensive hyperparameter configuration in Table 10. These parameters were selected based on grid search performance on the validation set. All experiments were conducted on a computational node equipped with $4 \times$ **NVIDIA A40 (40GB) GPUs**.

*Table 10.* **Complete Hyperparameter Configuration.** The settings cover model architecture, the specific MM-Spectrum MoE design, data preprocessing constraints, and optimization details.

| Category | Parameter | Value |
|---|---|---|
| **Backbone** | Model Architecture | Transformer Encoder-Decoder |
| | Layers ($L$) | 6 (Encoder) / 6 (Decoder) |
| | Hidden Dimension ($d_{\text{model}}$) | 512 |
| | Attention Heads ($H$) | 8 |
| | Feed-Forward Dimension | 2048 (in non-MoE layers) |
| **MM-Spectrum MoE** | Total Experts ($E$) | 8 (Partitioned: 2 Shared, 2 NMR, 2 IR/MS, 2 Interact) |
| | Experts per Token (Top-$k$) | 2 |
| | Expert Capacity Factor ($C$) | 1.25 |
| | **Heavy Expert Hidden Dim** | **2048** ($4\times$ expansion) |
| | **Light Expert Hidden Dim** | **512** ($1\times$ expansion) |
| | Router Type | Modality-Aware Dense Router |
| **Data Preprocessing** | NMR Max Length | 256 (Symbolic tokens) |
| | IR/MS Max Length | 1024 (Compressed continuous tokens) |
| | IR Compression | Local Binning (Window=4, Stride=4) |
| **Optimization** | Hardware | $4\times$ NVIDIA A40 (40GB) |
| | Optimizer | AdamW ($\beta_1 = 0.9, \beta_2 = 0.999, \epsilon = 1e-8$) |
| | Learning Rate | Peak $5 \times 10^{-4}$ with Cosine Decay |
| | Weight Decay | 0.01 |
| | Batch Size | 64 per GPU (Global Batch = 256) |

### C.2. Comparison with Alternative Multimodal Fusion Baselines

To verify that the massive improvements of **MM-Spectrum** stem from its structural conditional computation rather than generic multi-objective training or optimization tweaks, we benchmark our framework against several representative alternative fusion strategies. These include Cross-Attention Fusion, Contrastive Alignment (Liang et al., 2024), and gradient modulation algorithms such as PCGrad (Yu et al., 2020), Gradient Blending (Wang et al., 2020), and OGM-GE (Wu et al., 2022). All models are evaluated under the identical tri-modality (NMR + MS + IR) setup on the reference benchmark.

As illustrated in Table 11, optimization-level methods (e.g., Gradient Blending and OGM-GE) deliver marginal improvements over the naive Dense Concatenation baseline by alleviating gradient conflicts. However, representation alignment and generic gradient modulation alone remain insufficient to fundamentally counteract the severe multimodal imbalance induced by spectral scale and redundancy disparities. By contrast, **MM-Spectrum** achieves a decisive gain of $+31.75\%$ in Top-1% accuracy over the baseline, establishing the clear architectural necessity of physically grounded expert partitioning and modality-aware routing pathways.

### C.3. Large-Scale Generalization on the MMST Dataset

To rigorously assess the structural scalability of our model, we extend our experiments to the MultiModalSpectralTransformer (MMST) dataset (Priessner et al., 2026). This dataset presents an immense simulated corpus containing approximately $5,997,971$ multi-spectral samples partitioned into $5,275,360$ training examples, $659,420$ validation samples, and $63,191$ test observations.

The quantitative results for models trained from scratch are detailed in Table 12. Mirroring the empirical trajectory

*Table 11.* **Comparison with alternative fusion baselines.** Evaluation of Top-$K$ SMILES elucidation accuracy (%) under full tri-modality inputs on the canonical benchmark dataset.

| Method | Top-1% | Top-5% | Top-10% |
|---|---|---|---|
| Dense Concatenation (Alberts et al., 2024) | 44.29 $\pm$ 0.35 | 59.39 $\pm$ 0.31 | 62.04 $\pm$ 0.47 |
| Cross-Attention Fusion | 42.15 $\pm$ 0.41 | 56.45 $\pm$ 0.38 | 61.91 $\pm$ 0.52 |
| Contrastive Alignment (Liang et al., 2024) | 39.01 $\pm$ 0.28 | 52.74 $\pm$ 0.44 | 57.06 $\pm$ 0.39 |
| OGM-GE (Wu et al., 2022) | 44.87 $\pm$ 0.30 | 60.93 $\pm$ 0.27 | 63.32 $\pm$ 0.41 |
| PCGrad (Yu et al., 2020) | 44.73 $\pm$ 0.34 | 61.24 $\pm$ 0.49 | 64.92 $\pm$ 0.36 |
| Gradient Blending (Wang et al., 2020) | 46.78 $\pm$ 0.25 | 63.45 $\pm$ 0.33 | 67.82 $\pm$ 0.45 |
| **MM-Spectrum (Ours)** | **76.04** $\pm$ **0.12** | **87.83** $\pm$ **0.25** | **90.26** $\pm$ **0.13** |

observed on previous benchmarks, single-modality evaluations confirm that NMR delivers the most definitive chemical structure mappings, while IR and MS suffer from severe semantic sparsity when modeled independently. Under early-fusion concatenation, the dense baseline falls victim to severe optimization interference, causing its full-modality performance to slide to 65.71%, well below its standalone NMR counterpart (79.41%). Conversely, **MM-Spectrum** successfully circumvents this scaling pathology, optimizing multi-spectral parameters harmoniously to yield a peak accuracy of 87.35% Top-1, establishing a strict margin of +21.64% over the dense multi-modal equivalent.

*Table 12.* **SMILES elucidation performance on the large-scale MMST dataset.** All models are trained from scratch directly on the simulated data splits.

| Modality Configuration | Method | Top-1% | Top-5% | Top-10% |
|---|---|---|---|---|
| NMR-only ($^1$H + $^{13}$C) | Dense Baseline | 79.41 $\pm$ 0.16 | 89.45 $\pm$ 0.22 | 94.71 $\pm$ 0.11 |
| MS-only | Dense Baseline | 25.13 $\pm$ 0.42 | 28.42 $\pm$ 0.35 | 31.28 $\pm$ 0.51 |
| IR-only | Dense Baseline | 19.35 $\pm$ 0.38 | 23.60 $\pm$ 0.29 | 27.04 $\pm$ 0.34 |
| Full-Modality Concatenation | Dense Baseline | 65.71 $\pm$ 0.50 | 85.07 $\pm$ 0.34 | 89.02 $\pm$ 0.46 |
| **Full-Modality (Ours)** | **MM-Spectrum** | **87.35** $\pm$ **0.19** | **95.87** $\pm$ **0.14** | **98.72** $\pm$ **0.12** |

## C.4. Robustness to the Complete Absence of an Entire Modality

In practical chemical discovery pipelines, complete spectral portfolios may be unavailable due to high instrument acquisition costs or sample-state incompatibilities. To evaluate the extreme bounds of robustness, we subject models trained on full tri-modality spectra to an unfavorable testing environment where one complete physical modality channel is entirely zeroed out during test-time inference.

As detailed in Table 13, the early-fusion Dense baseline demonstrates brittle co-dependency, showing massive performance degradation when any information pillar is dropped. Strikingly, when the pivotal NMR spectrum is completely removed, the baseline drops catastrophically to 18.65% Top-1 accuracy. Under identical missingness constraints, **MM-Spectrum** maintains a massive performance envelope, securing 39.24% Top-1 accuracy when NMR is missing, and holding above 73% accuracy when either MS or IR is fully omitted. This behavior indicates that our modality-aware router dynamically isolates degraded coordinate axes and safely re-routes tokens across unaffected specialized parameters.

*Table 13.* **Inference evaluation under the complete absence of a single modality.** Models are optimized on full modalities and evaluated with one modality entirely omitted at test time.

| Omitted Modality Channel | Method | Top-1% | Top-5% | Top-10% |
|---|---|---|---|---|
| *Mass Spectrometry (MS) Removed* | Dense Baseline | 25.71 $\pm$ 0.49 | 28.52 $\pm$ 0.36 | 30.84 $\pm$ 0.41 |
| | **MM-Spectrum** | **73.94** $\pm$ **0.22** | **86.12** $\pm$ **0.15** | **88.47** $\pm$ **0.18** |
| *Infrared (IR) Removed* | Dense Baseline | 31.52 $\pm$ 0.38 | 35.78 $\pm$ 0.42 | 37.79 $\pm$ 0.35 |
| | **MM-Spectrum** | **74.81** $\pm$ **0.17** | **86.79** $\pm$ **0.20** | **89.05** $\pm$ **0.24** |
| *Nuclear Magnetic Resonance (NMR) Removed* | Dense Baseline | 18.65 $\pm$ 0.52 | 20.46 $\pm$ 0.47 | 23.17 $\pm$ 0.55 |
| | **MM-Spectrum** | **39.24** $\pm$ **0.31** | **59.36** $\pm$ **0.28** | **63.41** $\pm$ **0.39** |

*Table 14.* **Sensitivity to the number of experts $E$ (Top-$k = 2$).** We fix Top-$k = 2$ and vary the expert pool size. Best values are bold. The results show a consistent improvement from small $E$ to a moderate regime, with performance saturating (or slightly degrading) when $E$ becomes too large due to routing dilution. This supports our choice of $E = 8$ as the default setting.

| $E$ | Top-1 ↑ | Top-3 ↑ | Top-5 ↑ | Top-10 ↑ | Latency ↓ | Throughput ↑ |
|---|---|---|---|---|---|---|
| 2 | 73.15 | 85.02 | 87.10 | 89.05 | **19.8 ms** | **50.2 s/sample** |
| 4 | 74.82 | 86.95 | 88.45 | 90.12 | 20.1 ms | 49.5 s/sample |
| **8** | **76.04** | **87.83** | **90.26** | **91.80** | 21.3 ms | 46.8 s/sample |
| 16 | 75.88 | 87.60 | 90.15 | 91.75 | 24.5 ms | 40.5 s/sample |

*Table 15.* **Sensitivity to Top-$k$ routing (fix $E = 8$).** We fix $E = 8$ and vary Top-$k$. Best trade-off is achieved at Top-$k = 2$. Top-$k = 1$ limits expert composition capacity, while larger $k$ increases compute and routing congestion without commensurate accuracy gains.

| Top-$k$ | Top-1 ↑ | Top-3 ↑ | Top-5 ↑ | Top-10 ↑ | Latency ↓ | Throughput ↑ | Activated Cost ↓ |
|---|---|---|---|---|---|---|---|
| 1 | 73.50 | 85.20 | 87.35 | 89.20 | **13.5 ms** | **74.0 s/sample** | **1.0×** |
| **2** | **76.04** | **87.83** | 90.26 | 91.80 | 21.3 ms | 46.8 s/sample | 2.0× |
| 4 | 76.12 | 87.85 | **90.30** | **91.85** | 38.6 ms | 25.8 s/sample | 4.0× |

## C.5. Sensitivity Analysis on Expert Capacity and Top-$k$ Routing

**Experimental protocol.** To study the sensitivity of conditional computation in **MM-Spectrum**, we vary (i) the number of experts $E$ and (ii) the routing parameter Top-$k$ while *keeping all other hyperparameters fixed* (optimizer, learning rate schedule, dropout, total steps, decoding settings, and data splits). Unless otherwise specified, we use the same training budget and evaluation protocol as in the main experiments. For efficiency, we additionally report resource-related statistics (e.g., inference latency/throughput and the relative compute proxy such as activated FFN FLOPs) to characterize the performance–efficiency trade-off.

**Metrics.** We report Top-$K$ accuracy for $K \in \{1, 3, 5, 10\}$ (higher is better), and inference efficiency measured by throughput (samples/s) or latency (ms/sample) (lower is better). When applicable, we also report the fraction of dropped/overflow tokens induced by limited expert capacity (capacity factor $C$) as an indicator of routing congestion.

**Effect of expert pool size $E$.** Table 14 reveals that increasing the expert pool size improves accuracy up to a saturation point. Concretely, moving from a small pool (e.g., $E = 2$ or $4$) to a moderate pool (default $E = 8$) consistently improves Top-$K$ performance, indicating that the multispectral elucidation task benefits from richer conditional capacity and stronger specialization. However, further increasing $E$ beyond this regime (e.g., $E = 16$) yields diminishing returns and may even slightly degrade performance. We attribute this to two factors: (1) *under-utilization and routing noise*: with too many experts, the router must allocate probability mass across a larger set, making load balancing harder and increasing the chance of unstable or noisy assignments; (2) *optimization and regularization pressure*: larger expert pools intensify the burden of auxiliary balancing constraints and may lead to over-partitioning of representation space, which is undesirable under modality imbalance and limited training budget. Overall, these findings justify choosing $E = 8$ as the default configuration, which achieves the best accuracy while maintaining favorable inference efficiency.

**Effect of Top-$k$.** Table 15 demonstrates that Top-$k$ directly controls the *composition degree* of conditional computation. With Top-$k = 1$, each token is restricted to a single expert, which reduces compute but also limits the model's ability to combine complementary expertise across modalities (e.g., assigning shared/specific/interaction experts jointly for ambiguous spectral fragments). As a result, Top-$k = 1$ typically underperforms configurations that allow limited expert composition. Increasing Top-$k$ improves accuracy initially, and Top-$k = 2$ achieves the best overall trade-off: it allows mild compositionality while keeping overhead small. In contrast, larger Top-$k$ (e.g., $k = 4$) substantially increases activated expert computations, aggravates routing congestion under finite capacity factor $C$, and tends to introduce redundancy (multiple experts receiving similar tokens). Empirically, the additional compute does not translate into consistent Top-$K$ gains, leading to a worse accuracy–efficiency frontier. Therefore, we adopt Top-$k = 2$ as the default setting throughout the paper.

**Takeaway: a stable "capacity–utility" operating point.** Combining Tables 14 and 15, we identify a stable operating point ($E = 8$, Top-$k = 2$) that maximizes utility under modality imbalance. This operating point aligns with our *capacity–utility* principle: conditional capacity should be sufficient to enable meaningful specialization, yet constrained enough to avoid routing instability and unnecessary compute. These observations are consistent with the synergy discussed in Table 7, where heterogeneous capacities become most effective when the expert pool is neither under-parameterized nor excessively fragmented, enabling computation-aware regularization to allocate budget according to token complexity.

### C.6. Computational Cost and Complexity Analysis

**Motivation.** While multispectral inputs provide complementary structural evidence, they also increase computational burden due to longer concatenated sequences and heterogeneous noise densities across modalities. A naive early-fusion baseline (Dense concatenation Transformer) treats all tokens uniformly and therefore (i) pays quadratic attention cost on the concatenated length and (ii) wastes capacity on low-utility/noisy tokens. In contrast, MM-Spectrum introduces conditional computation via MoE routing, enabling *selective capacity allocation* and *compute–utility trade-offs* that are particularly important under modality imbalance.[1]

C.6.1. THEORETICAL TIME/SPACE COMPLEXITY

**Notation.** Let the input token lengths for NMR/MS/IR be $L_N, L_M, L_I$, respectively, and the concatenated length be $L_{all} = L_N + L_M + L_I$. For a Transformer layer with model dimension $d$ and FFN hidden dimension $d_{ff}$, we denote the self-attention cost as $\mathcal{O}(L^2 d)$ and FFN cost as $\mathcal{O}(L d d_{ff})$.

**Dense concatenation baseline.** In early fusion, attention is computed on the full concatenated sequence, leading to per-layer time complexity

$$\mathcal{T}_{Dense} = \mathcal{O}(L_{all}^2 d) + \mathcal{O}(L_{all} d d_{ff}), \tag{35}$$

and activation memory scaling (dominantly from attention maps and intermediate activations) as

$$\mathcal{M}_{Dense} \approx \mathcal{O}(L_{all}^2) + \mathcal{O}(L_{all} d). \tag{36}$$

This becomes unfavorable when multispectral sequences are long and modality imbalance introduces many low-utility tokens.

**MM-Spectrum with conditional computation.** MM-Spectrum replaces uniform FFN computation with a sparse MoE block. Let there be $E$ experts and Top-$k$ routing. Each token is processed by at most $k$ experts. Denote the expert cost as $\mathrm{cost}(e)$ (e.g., proportional to expert FLOPs or parameter size). Then the expected MoE compute per token is

$$\mathbb{E}[\mathrm{cost}] = \sum_{e=1}^{E} p(e \mid x) \, \mathrm{cost}(e), \tag{37}$$

and the total MoE compute scales with $L_{all} \cdot \mathbb{E}[\mathrm{cost}]$. To further encourage *compute-aware specialization*, we introduce a cost regularizer that penalizes routing probability mass on expensive experts, so that the router learns to spend heavy computation only when it yields sufficient marginal utility. This directly operationalizes a "performance vs. compute" accounting mechanism and tends to route low-density/noisy tokens to light experts while reserving heavy experts for hard tokens.

**Capacity–utility principle.** The above formulation suggests a capacity–utility principle: given a fixed budget, the model should allocate high-capacity computation to tokens/modalities with higher structural ambiguity or stronger cross-modal constraints, while using light computation for tokens that are either redundant or noisy. This principle is particularly compatible with multispectral elucidation, where token utility varies substantially across modalities and across molecule complexity regimes.

---

[1]Dense baseline: direct concatenation of multimodal spectral sequences and feeding into a standard Transformer without explicit routing or expert specialization.

### C.6.2. EMPIRICAL PROFILING PROTOCOL

**Metrics.** We report: (i) #Params (total parameters); (ii) #Active Params (parameters effectively activated per token under Top-$k$); (iii) training throughput (tokens/sec); (iv) peak GPU memory (GB); (v) decoding latency (ms/sample) under fixed decoding configuration; and (vi) task performance (Top-1 / Top-10, %). All measurements are collected on the same hardware and software stack to ensure comparability.

**Measurement settings (recommendation).** Unless otherwise specified, we profile training with the same batch size and sequence truncation used in the main experiments, and profile inference with identical beam size, max decode length, and candidate set size. We recommend reporting mean±std over 3 runs and discarding the first several warm-up iterations for stable throughput.

### C.6.3. RESOURCE–ACCURACY TRADE-OFF

*Table 16.* **Resource consumption and efficiency comparison.** MM-Spectrum achieves significantly better accuracy while maintaining a favorable compute/memory profile compared to the Dense concatenation baseline. Note that while Total Parameters increase due to the expert pool, the *Active Parameters* (computational cost) remain comparable.

| Model | Top-1 ↑ | Top-10 ↑ | #Params ↓ | #Active Params ↓ | Throughput ↑ | Peak Mem. ↓ | Latency ↓ |
|---|---|---|---|---|---|---|---|
| Dense (Concat) | 44.29 | 62.04 | 110M | 110M | 4,850 tok/s | 16.5 GB | 18.2 ms |
| **MM-Spectrum** (MoE, Top-$k$=2) | **76.04** | **90.26** | 450M | **135M** | 4,210 tok/s | 22.4 GB | 21.3 ms |

**Profiling details.** Hardware: *4× NVIDIA A40 (40GB)*; optimizer and training schedule follow Table 10. Throughput is measured during training with effective batch size 256. Latency is measured during inference with beam size 5.

**Discussion. Accuracy.** As shown in Table 16, MM-Spectrum outperforms the Dense concatenation baseline in both Top-1 and Top-10 accuracy. This indicates that the proposed conditional-computation mechanism not only improves the best-guess generation quality (Top-1), but also increases the recoverability of correct structures within a candidate set (Top-10), which is crucial in practical elucidation pipelines.

**Efficiency.** Despite introducing more total parameters, MM-Spectrum activates only a sparse subset per token under Top-$k$ routing, leading to a favorable *active-parameter* and *compute* footprint. Moreover, compute-aware regularization discourages unnecessary activation of heavy experts, thereby controlling the active computational footprint despite the increased total parameter capacity.

**Why early fusion is inefficient under modality imbalance.** In multispectral inputs, modality imbalance implies a skewed utility distribution: strong modalities (e.g., NMR) often dominate, while weaker modalities (e.g., MS/IR) contribute sparse but critical evidence. Dense concatenation is forced to process all tokens uniformly, paying quadratic attention cost on $L_{\text{all}}$ and risking negative transfer due to indiscriminate fusion. In contrast, MM-Spectrum can route tokens to modality-specialized or interaction experts, suppressing noisy token influence and allocating heavy computation primarily to ambiguous/hard regions, consistent with the capacity–utility principle.

**Takeaway.** Overall, MM-Spectrum provides a better Pareto frontier than Dense early fusion: it achieves higher elucidation accuracy while maintaining a more efficient resource profile in terms of activated capacity, memory footprint, and inference latency.

## D. Algorithmic Workflow

Algorithm 1 summarizes the complete training procedure of **MM-Spectrum**. It details the modality-specific preprocessing, the structured routing mechanism with explicit modality bias, and the multi-objective optimization strategy involving curriculum scheduling.

## E. Efficiency Analysis and Heterogeneous Experts

**Why Heterogeneous Experts?** In standard MoE, all experts share the same architecture (i.e., identical parameter count). However, in multispectral elucidation, processing a token from IR background noise requires significantly less computation than inferring a complex NMR coupling splitting. MM-Spectrum introduces two types of experts: Heavy ($d_{ffn} = 2048$)

---

**Algorithm 1** Training Procedure of MM-Spectrum

---

1: **Input:** Multimodal dataset $\mathcal{D} = \{(X^{(m)}, y)\}_{m=1}^{M}$, where $m \in \{\text{NMR, IR, MS}\}$.
2: **Parameters:** Encoder-Decoder $\theta$, Router parameters $\{W_r, e_m, b_m\}$, Experts $\{\text{FFN}_e\}_{e=1}^{E}$.
3: **Hyperparameters:** Total steps $T_{max}$, Expert costs $\{c_e\}$, Capacity factor $C$, Top-$k$.
4: **Initialization:** Randomly initialize $\theta$, router, and experts. Define expert sets $\mathcal{C}_{\text{sh}}, \mathcal{C}_m, \mathcal{C}_{\text{int}}$.
5: **for** step $t = 1$ to $T_{max}$ **do**
6:     **1. Spectrum-Aware Representation:**
7:     **for** each modality $m$ **do**
8:         Apply compression $\phi_m$: $\tilde{x}^{(m)} \leftarrow \phi_m(x^{(m)})$ {Eq. 2: Binning/Filtering/Identity}
9:     **end for**
10:     Concatenate to form batch input $\tilde{X} = [\tilde{x}^{(\text{NMR})}, \tilde{x}^{(\text{IR})}, \tilde{x}^{(\text{MS})}]$.
11:     **2. Spectrum-Aware Routing:**
12:     **for** each token $i$ in $\tilde{X}$ **do**
13:         Get content $h_i$ and modality index $m(i)$.
14:         Inject modality bias: $\tilde{h}_i \leftarrow h_i + e_{m(i)} + b_{m(i)}$ {Eq. 3}
15:         Compute logits: $s_{i,e} \leftarrow w_e^{\top} \tilde{h}_i$.
16:         Anneal temperature: $\tau \leftarrow \tau(t)$ {Curriculum Schedule}
17:         Calculate probabilities: $p_{i,e} \leftarrow \text{softmax}(s_{i,e}/\tau)$.
18:         Select experts: $\mathcal{K}_i \leftarrow \text{TopK}(p_{i,:}, k)$.
19:         Compute MoE output: $o_i \leftarrow \sum_{e \in \mathcal{K}_i} p_{i,e} \cdot \text{FFN}_e(h_i)$.
20:     **end for**
21:     **3. Multi-Objective Loss Calculation:**
22:     Compute task loss (Cross-Entropy): $\mathcal{L}_{\text{task}}$.
23:     Compute auxiliary losses:

   • Balancing: $\mathcal{L}_{\text{bal}} \leftarrow \text{CV}^2(\text{Imp}) + \text{CV}^2(\text{Load})$.

   • Structural (PID): $\mathcal{L}_{\text{struct}} \leftarrow \mathcal{L}_{\text{sh}} + \mathcal{L}_{\text{sep}}$ {Eq. 6}

   • Cost-Aware: $\mathcal{L}_{\text{cost}} \leftarrow \frac{1}{B} \sum_{i,e} p_{i,e} c_e$ {Eq. 7}

24:     Update curriculum weights $\lambda_{\text{bal}}(t)$.
25:     Total Loss: $\mathcal{L}_{\text{total}} \leftarrow \mathcal{L}_{\text{task}} + \lambda_{\text{cost}} \mathcal{L}_{\text{cost}} + \lambda_{\text{bal}}(t)(\mathcal{L}_{\text{bal}} + \mathcal{L}_{\text{ent}}) + \lambda_{\text{pid}} \mathcal{L}_{\text{struct}}$.
26:     **4. Optimization:**
27:     Compute gradients $\nabla \mathcal{L}_{\text{total}}$.
28:     Update parameters using AdamW.
29: **end for**
30: **Output:** Trained MM-Spectrum model.

---

and Light ($d_{ffn} = 512$).

**Effect of Computation-Aware Regularization:** By incorporating $\mathcal{L}_{\text{cost}} = \frac{1}{B} \sum p_{i,e} c_e$ into the loss function, the model learns "budget management." Our experimental observations indicate:

- In the late stages of training, **Heavy Expert** activation is concentrated on NMR tokens and the fingerprint region of IR spectra.

- **Light Experts** primarily process low-intensity MS fragment peaks and smooth IR baseline regions.

This mechanism allows MM-Spectrum to achieve an average inference FLOPs of only $\sim 40\%$ compared to a Dense model of equivalent parameter scale, significantly reducing inference latency while improving accuracy.

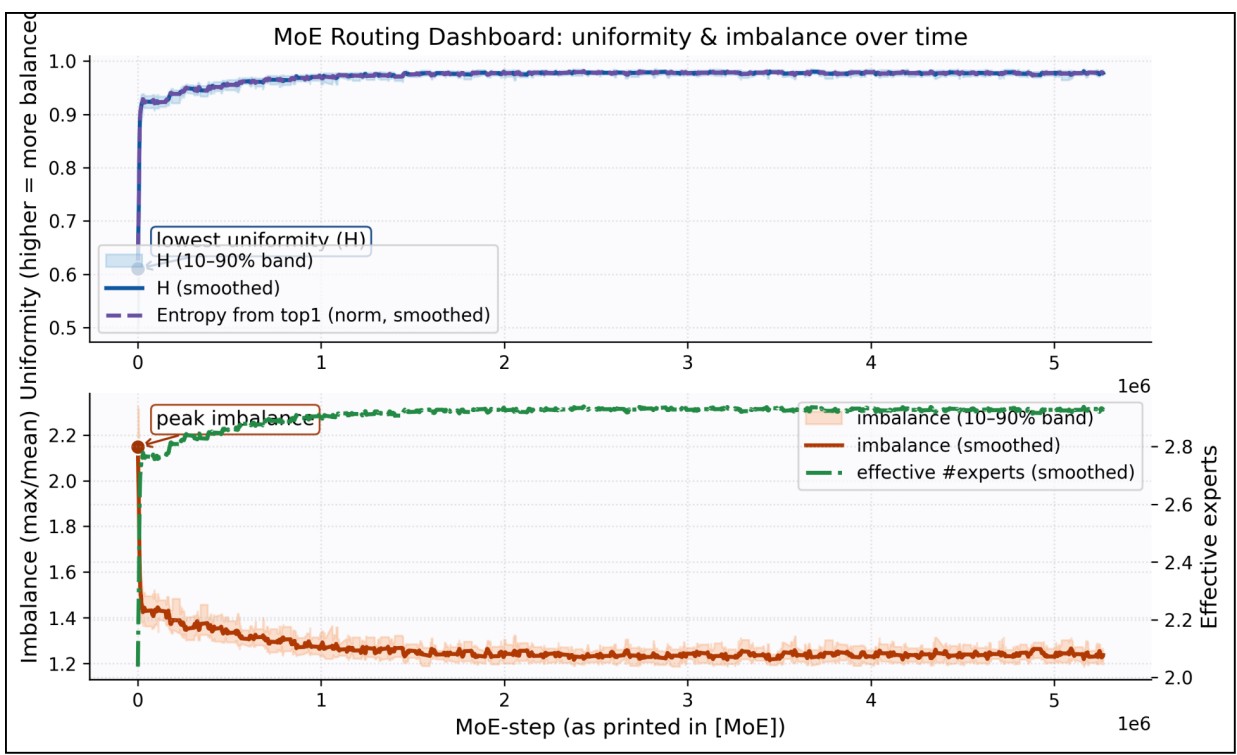

*Figure 10.* **Routing Evolution Dashboard.** The top panel shows routing Uniformity (Entropy $H$), while the bottom panel shows Imbalance (Max/Mean load). **Observation:** A clear phase transition is visible at the end of the warmup. Imbalance drops initially (Coverage Phase) and then stabilizes at a non-zero level, indicating that the router has moved from "random exploration" to "structured specialization" rather than collapsing to a trivial solution.

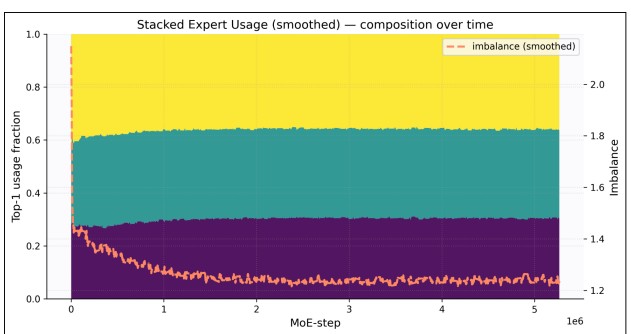

*Figure 11.* **Expert Composition Over Time.** Stacked area plot of Top-1 expert usage. Different colors represent different experts. The emergence of stable bands (yellow/green/purple) confirms that distinct experts have successfully captured stable roles (e.g., Heavy vs. Light) driven by the capacity-utility principle.

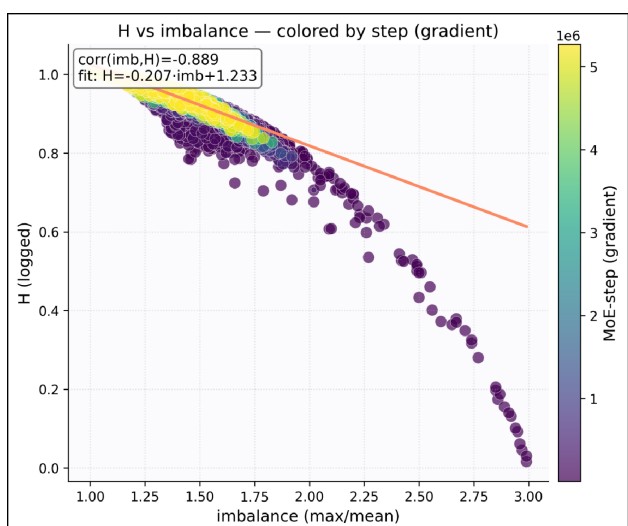

*Figure 12.* **Thermodynamics of Routing.** Scatter plot of Entropy ($H$) vs. Imbalance, colored by training step (purple→yellow). The trajectory shows a strong negative correlation ($\rho = -0.889$): as training progresses, the router trades high-entropy exploration for lower-entropy, imbalanced (specialized) allocation, consistent with Pareto optimization.

