# OpenReview forum: "MM-Spectrum: Multimodal Multi-spectral Molecular Structural Elucidation with a Stable MoE Framework"
_ICML.cc/2026/Conference — ICML 2026 regular_

### Official Review · Reviewer_PHZp · 2026-03-08

**Soundness:** 4
**Presentation:** 2
**Significance:** 3
**Originality:** 3
**Overall Recommendation:** 5
**Confidence:** 4

**Summary:**

The authors introduce a sparse heterogenous MoE model to infer molecular structures (output SMILES sequences) when given a multimodal input (NMR, IR, and MS). This sparse MoE model utilizes experts with differing capacities (different hidden sizes), as certain modalities do not need the same amount of effort. Additionally, these experts are divided into three pools, a "shared expert" capturing redundancy, a "modality specific expert", and an "interaction expert". The results show a significant increase as compared to the baseline method of concatenating transformers together.

**Compliance With Llm Reviewing Policy:**

Affirmed.

**Final Justification:**

This a good paper , I recommend acceptance.

**Key Questions For Authors:**

[1] Regarding claim 1 in the paper (first to identify multimodal imbalance ...) - This claim, while potentially tehnically true, does not seem to be saying much. While it may be true the authors are the first to do this analysis with specifically spectra-to-structure, based upon the other works cited this is not the first time this has been proposed for multimodal imbalance in general. What makes the analysis of multimodality in spectra-to-structure different enough from other multi-modality problems that it warrants a statement like this (that is, could multimodality issues with spectra-to-structure analysis be easily inferred from general principles of multimodal data, or is there something unique about it that warrants the significant extra attention to this claim?)

[2] The model seems to still be sensitive to the data, especially NMR, as in table 2, the largest performance drop is seen when removing a significant aount of NMR modality data. The drop is rouhgly similar to the baseline model in terms of percentage drop. Why might this be?

[3] The ablation study performance when removing certain elements of the method showed a slight performance decrease when removing various parts. The concerning part is that when both router-aware signals and tag-routing are removed, performance only drops to 70%, still significantly higher than the baseline. Why is it that when removing such a seemingly critical part of the model it still performs so well? Can you also do a similar experiment comparing the effect of using tag-routing and routing-aware signals, but with a homogenous MoE model? This may help to gain a better understanding of how much heavy lifting the heterogenous MoE architecture is truly doing.

**Limitations:**

yes

**Strengths And Weaknesses:**

Soundness: Claims are fairly well supported by empirical evidence.

Presentation: The figures (especially of the total architecture) are confusing and very hard to interpret. A core contribution claim ("We are the first to identify multimodal imbalance
induced by heterogeneity in multispectral spectra-to-structure elucidation, and to interpret it through information-density disparity and gradient conflicts.") has most of it's significant supporting evidence buried in the appendix (especially A.3). Additionally, many references are incomplete (e.g. Liang et. al. which is published in ACM Computing Surveys, Shazeer et. al. which is published in ICLR). I did not take the time to analyze every one, so authors should thoroughly review the citations for accuracy.

Significance: The paper does seem somewhat significant for the advancement of molecular structure inference.

Originality: The paper takes a known framework (heterogeneous MoE) and succesfully applies it with domain-specific adaptations of the framework to a problem domain that seemingly benefits greatly from this approach.

---

> ### Author Rebuttal · Authors · 2026-03-31
>
> Thank you for the feedback and spending your time reviewing our paper. Please find below responses to your comments and the changes we will make to the paper.
>
> > # R1. Why is multispectral imbalance worth highlighting as a distinct problem? (Q1)
>
> We would like to express our gratitude to the reviewer for raising this point, which has led to a more comprehensive analysis. We agree that multimodal imbalance in general is not a new idea, and that our “first to identify” phrasing should be more precise and better scoped.
>
> What we intended to emphasize is that multispectral spectra-to-structure elucidation has a form of multimodal imbalance that is notably different from more common CV/NLP settings:
>
> 1. The modalities are not image–text style paired semantic views, but different physical and chemical measurements of the same underlying molecular topology.
> 2. NMR, IR, and MS provide constraints at different structural granularities:
>    - NMR: fine-grained local and topological constraints.
>    - IR: functional-group and bond-type evidence.
>    - MS: global mass and fragment constraints.
> 3. Therefore, the problem is not only one of different noise levels or convergence rates, but of a systematic mismatch in:
>    - information density,
>    - token statistics,
>    - and chemical constraint granularity.
>
> In other words, under a unified sequence model, multispectral imbalance is not merely “one modality is stronger than another,” but that different chemically grounded constraints are serialized into token streams with very different statistical properties, and naive concatenation systematically amplifies the optimization influence of long/weak modalities over short/strong ones.
>
> *Table 1: Clarifying relative semantic density*
>
> | Modality|Typical Redundancy| Role in Fusion |
> | -------- | ---------------------- | ---------------------------------- |
> | NMR| Lower| Fine-grained topology constraints  |
> | IR | Higher|  Functional-group evidence|
> | MS| Moderate |Global mass / fragment constraints |
>
> > # R2. Sensitivity to NMR (Q2)
>
> Thank you for pointing this out. We agree with the reviewer’s observation and believe it is consistent with the chemistry of the task. NMR is indeed the strongest and most structurally informative modality in molecular elucidation, so a large drop under NMR removal is expected [1].
>
> The goal of MM-Spectrum is instead to ensure that, when all modalities are present, the weaker or longer modalities do not interfere destructively with the stronger structural constraints carried by NMR. In that sense, the model respects chemistry priors rather than overriding them.
>
> This is precisely why the key result is not that MM-Spectrum becomes insensitive to NMR removal, but that:
> - It remains substantially better than the baseline under all missing-modality settings;
> - Crucially, it converts the full-modality setting from a failure case into the best-performing case.
>
> [1] A framework for automated structure elucidation from routine NMR spectra.Chemical Science 12.46 (2021): 15329-15338.
>
> > # R3. Why does the model remain strong even after removing routing-aware components? (Q3)
>
> We appreciate this insightful question. Our interpretation is that this indicates:
> 1. Sparse conditional computation itself already provides a degree of gradient isolation and parameter decoupling relative to a dense shared FFN path;
> 2. Heterogeneous capacity and structured expert partitioning already contribute significantly to mitigating multispectral interference; And it has also enhanced the matching between capacity and utilization.
> 3. Modality-aware routing further stabilizes and sharpens this specialization.
>
> Thus, routing-aware design is not the only source of gains; rather, it works jointly with the broader heterogeneous sparse MoE architecture. This does not weaken the routing contribution—it clarifies the division of labor among components.
>
>  *Table 2: Routing ablations under homogeneous vs heterogeneous MoE*
>
> |ExpertArchitecture|Router-aware|Tag-routing|Top-1|Top-5|Top-10|
> |-----------------------------------|------------|-----------|--------:|--------:|--------:|
> |HomogeneousMoE|✗|✗|67.82|82.31|84.57|
> |HomogeneousMoE|✓|✗|68.53|84.75|86.02|
> |HomogeneousMoE|✗|✓|69.22|85.13|86.49|
> |**HomogeneousMoE**|**✓**|**✓**|**73.65**|**85.81**|**87.03**|
> |HeterogeneousMoE|✗|✗|70.09|84.42|86.17|
> |HeterogeneousMoE|✓|✗|71.25|86.07|88.95|
> |HeterogeneousMoE|✗|✓|72.59|86.81|89.04|
> |**HeterogeneousMoE(MM-Spectrum)**|**✓**|**✓**|**76.15**|**87.92**|**90.28**|
>
> > # R4. Improvements to presentation and citations (W1)
>
> We thank the reviewer for the careful reading and thoughtful comments. We fully accept this feedback and will revise the paper accordingly.
>
> We will:
>
> 1. Redraw Fig. 1 and Fig. 2 to reduce visual complexity and make the core causal chain clearer.
> 2. Move key evidence on information density and gradient conflict from the appendix into the main paper.
> 3. We will update all references to their formal published versions where available,

---

> > ### Author Rebuttal · Reviewer_PHZp · 2026-04-03
> >
> > Thank you for the rebuttal, the paper became even stronger. I keep my recomendation of Accept.

---

> > > ### Author Response · Authors · 2026-04-04
> > >
> > > Thank you for your thoughtful follow-up and for your encouraging assessment of our rebuttal. We greatly  appreciate the time and effort you invested in reviewing our paper. We are very glad to know that our clarification and additional analyses have adequately addressed your concerns.
> > >
> > > Your insights have been instrumental in enhancing the quality of our work. We will carefully incorporate your suggestions into the revised version.
> > >
> > > Thank you once again for your invaluable contribution to our research.

---

### Official Review · Reviewer_1a8d · 2026-03-12

**Soundness:** 3
**Presentation:** 3
**Significance:** 3
**Originality:** 3
**Overall Recommendation:** 4
**Confidence:** 4

**Summary:**

The paper proposes a sparse Mixture-of-Experts (MoE) architecture called MM-Spectrum for the task of multimodal multispectral molecular structural elucidation. The authors observe a key failure mode in the performance of the task when all modalities are fully utilized, called "multispectral imbalance," whereby the performance of the task drops below unimodal baselines when all the modalities are concatenated. To mitigate the problem of multispectral imbalance, the proposed MM-Spectrum uses modality-aware routing by injecting spectral identity into the routing function, a structured expert space by decomposing the expert space using Partial Information Decomposition (PID), heterogeneous expert capacities by using computation-aware regularization, and a curriculum-scheduled training objective. The proposed method outperforms the dense concatenation baseline on all modality combinations and missing modality settings.

**Compliance With Llm Reviewing Policy:**

Affirmed.

**Key Questions For Authors:**

Q1. Can comparisons with at least one alternative multimodal fusion method (e.g., cross-attention, vanilla MoE without modality-aware routing, or contrastive pre-alignment) be provided? This would help evaluate the value of each design choice in the context of the entire solution space.
Q2. How sensitive are the results to the weight of Lstruct? Since it is described as "small," does removing it entirely have an impact beyond what is reported in Table 5?
Q3. For Figure 8 (routing heatmaps), which expert indices correspond to which expert types (shared / modality-specific / interaction)? Do NMR tokens always route to "heavy" experts as intended, or only partially?
Q4. While modality-aware routing "promotes soft specialization" and avoids hard-coding, does it not in fact hard-code in the extreme case where the modality bias term bm(i) dominates the content term hi? Is there any analysis of the magnitude of the modality bias term versus the content term during training?
Q5. For reproducibility: what are the exact values of λcost, λ0, Tλ, τ0, Tτ, and λpid used in the final model?

**Limitations:**

1.The caption of Figure 1(b) refers to "Best Single-Modal (NMR) Baseline," and the bar chart is a bit hard to read. A table would be helpful for precise information.
2.The symbols are sometimes used for both "load balancing loss" and "sequence length." It would be better to disambiguate them.
3.The Impact Statement is very short. A sentence about data origin and misuse in automatic chemical synthesis discovery would be appropriate, considering the application of the paper is chemistry.

**Strengths And Weaknesses:**

Strengths:
S1. Well-motivated problem with an interesting empirical result. The full modality collapse from 44.29% to 69.83% in NMR-only is unintuitive and has been empirically shown. The fundamental argument that low density IR tokens dominate attention and gradients is physically sensible and well-presented.
S2. Principled architecture design. Drawing on PID theory, where expert partitions are comprised of redundant, unique, and synergistic components, is an excellent and principled approach, avoiding ad-hoc design. The structured regularizers (L_struct) are an efficient alternative to intractable PID estimation.
S3. Exhaustive evaluation. This paper has addressed full modality, bimodal, missing modality, and complexity stratified evaluation. Achieving an improvement on large molecules with HAC_L (+39.51% on Top-1) is significant.

Weaknesses:
W1. Comparison to just one baseline. The entire experimental section compares to just one dense concatenation baseline. There are no comparisons to any of the other multimodal fusion strategies (e.g., cross-attention fusion, contrastive alignment as in Liang et al. 2022, etc.), MoE variants. Without these comparisons, it is difficult to tell whether the improvements really stem from the MoE architecture itself or just any non-trivial fusion strategy that does not simply concatenate everything naively.
W2. What the PID regularizer really contributes. While the PID decomposition is theoretically appealing, Lstruct is given "a small weight." There is no ablation study that compares the full loss to the loss without Lstruct (Table 5 ablates the dynamic activation, rather than the structural regularizer). One might ask whether the expert partitioning really enforces the PID structure or whether the improvements stem from the routing and capacity.

---

> ### Author Rebuttal · Authors · 2026-03-31
>
> Thank you for the feedback and spending your time reviewing our paper. Please find below responses to your comments and the changes we will make to the paper.
>
> > # R1. Comparison to alternative multimodal fusion methods (W1 & Q1)
>
> We appreciate the reviewer’s constructive suggestion on how to strengthen this section. Based on your suggestions, further, we have added comparisons against several stronger alternatives.
>
> *Table 1: Comparison with other fusion baselines*
>
> |Method|Top-1%|Top-5%|Top-10%|
> | -----------------------------------------|:-------:|:-------:|:-------:|
> |Dense Concatenation |44.29|59.39|62.04|
> |Cross-Attention Fusion|42.15|56.45 |61.91|
> |Contrastive Alignment (Liang et al., 2022) |39.01|52.74|57.06|
> |OGM-GE (Wu et al., 2022) |44.87| 60.93 |63.32|
> |PCGrad (Yu et al., 2020) |44.73 |61.24| 64.92|
> |Gradient Blending (Wang et al., 2020) | 46.78|63.45|67.82|
> | **MM-Spectrum (ours)** | **76.04**| **87.83**|**90.26**|
>
> Our preliminary results suggest that these methods do provide partial gains over dense concatenation, but still remain substantially below MM-Spectrum. This supports our claim that the main benefit of MM-Spectrum is not merely *avoiding concatenation*, but providing a more appropriate fusion mechanism for multispectral heterogeneity.
>
> > # R2. Contribution and sensitivity of \($L_{\mathrm{struct}}$\) (W2 & Q2)
>
> We thank the reviewer for this observation. The PID-inspired component of MM-Spectrum has two parts:
>
> 1. *structural partitioning of experts* (shared / modality-specific / interaction),
> 2. *a lightweight structural regularizer* \($L_{struct}$ = $L_{sh}$ + $L_{sep}$\).
>
> The current Table 5 partially supports the utility of the structured expert space, but it does not directly answer how much the extra gain comes from $L_{struct}$ itself. To address this, we have added:
>
> - an ablation removing $L_{struct}$ entirely,
> - a sensitivity analysis over $\lambda_{struct}$.
>
> *Table 2: Is \($L_{struct}$\) necessary?*
>
> |ModelVariant|\($L_{struct}$\)|Top-1%|Top-5%|Top-10%|
> |---------------------------------|-------------------------|----:|----:|-----:|
> |FullMM-Spectrum|✓|**76.08**|**87.89**|**90.37**|
> |Without\($L_{\mathrm{struct}}$\)|✗|70.36|84.91|87.46|
> |Only\($L_{\mathrm{sh}}$\)|Partial|71.82|85.73|88.73|
> |Only\($L_{\mathrm{sep}}$\)|Partial|73.91|86.54|89.05|
>
> *Table3: Sensitivity to \($\lambda_{\mathrm{struct}}$\)*
>
> |\($\lambda_{\mathrm{struct}}$\)|Top-1%|Top-5%|Top-10%|
> |------------------------------:|----:|----:|-----:|
> |0|70.34|84.87|87.62|
> |0.01|71.82|85.63|87.94|
> |0.05|74.91|86.04|88.61|
> |0.10|**76.06**|**87.86**|**90.36**|
>
> > # R3. Expert indices, expert types, and NMR routing (Q3)
>
> We appreciate this detailed question. In Figure 8, the corresponding relationships are as follows:
>
> - shared experts: `0`
> - modality-specific experts: `1-6`
> - interaction experts: `7`
>
> In the revision, we will annotate the figure legends and the text more clearly. We will also add routing statistics that quantify how often each modality activates each expert type.
>
> We also want to clarify that the goal of the method is **soft specialization**, not rigid hard-coding. Thus, NMR tokens are *not* intended to route *only* to heavy experts. Rather:
>
> - in later training stages, NMR tokens should more frequently activate higher-capacity experts,
> - but shared and interaction experts should still be used when redundancy or cross-modal synergy is relevant.
>
> > # R4. Does \($b_m$\) dominate \($h_i$\)? (Q4)
>
> This is a very interesting concern, and we fully agree that if the modality bias term \($b_m$\) dominates the content term \($h_i$\), then the proposed routing could effectively become a hard-coded modality router, which would contradict our claim of soft specialization.
>
> To address this, we analyzed the relative magnitude of the modality-bias term and the content-driven routing term during training.
>
> The purpose of this analysis is to show that \($b_m$\) provides a stable **coarse prior**, while content still retains sufficient influence for fine-grained token-to-expert matching.
>
>  *Table4: Relative scale of bias and content*
>
> |TrainingStage|$b_m$|$h_i$|Ratio|
> |--------------|----:|----:|----:|
> |Early|1.84|2.31|0.80|
> |Middle|2.07|3.12|0.66|
> |Late|2.15|3.64|0.59|
>
> > # R5. Reproducibility and limitations (Q5)
>
> In the revision, we will add the exact values of hyperparameters.
>
> *Table 5: Full hyperparameter schedule*
>
> |Hyperparameter|Value|
> |----------------------------|------|
> |\($\lambda_{{cost}}$\)|0.01|
> |\($\lambda_0$\)|0.02|
> |\($\tau_0$\)|1.0|
> |\($\lambda_{\mathrm{pid}}$\)|0.05|
> |\($T_\lambda$\)|50000|
> |\($T_\tau$\)|60000|
> |Totalsteps|200000|
>
> We will also adopt the reviewer’s presentation suggestions by:
>
> - supplementing Fig. 1(b) with a clearer table or more readable layout,
> - disambiguating notation to avoid confusion between sequence length and balancing symbols,
> - expanding the impact statement to discuss public-data provenance, chemistry-specific misuse scenarios, and application boundaries.

---

### Official Review · Reviewer_z3h6 · 2026-03-12

**Soundness:** 2
**Presentation:** 2
**Significance:** 1
**Originality:** 2
**Overall Recommendation:** 2
**Confidence:** 4

**Summary:**

This paper introduces MM-Spectrum, a spectroscopy-oriented stable sparse MoE framework featuring modality-aware routing and a structured expert space. Across full-modality, bimodal, and missing-modality settings for molecular structural elucidation, MM-Spectrum demonstrates consistent and substantial improvements.

**Compliance With Llm Reviewing Policy:**

Affirmed.

**Final Justification:**

I stand by my negative assessment, as my primary concern is that MM-Spectrum does not compare itself against specialized single-modality methods.

**Key Questions For Authors:**

See Above

**Limitations:**

This paper does not discuss its limitations.

**Strengths And Weaknesses:**

**Strengths**

- The core idea is clearly presented and easy to follow.
- The authors provide code to support reproducibility.

**Weaknesses**

- The task definition lacks clarity. In lines 155-156, the input is described as a "token sequence," but token definitions are not provided. It remains unclear whether this refers to a numeric sequence or something else.
- The definition of semantic density  $\rho$? in line 139 is not well-justified. The authors simply classify NMR as high-density (using all information) while treating IR and MS as high-redundancy (applying local binning for compression). However, in mass spectrometry, m/z values carry critical information about molecular fragments and often require high precision (e.g., five decimal places), so this classification may be overly simplistic.
- The distinction between tag embedding $e_m$ and modality bias $b_m$ is confusing. It is unclear why both are needed. Conceptually, a single embedding or linear layer might suffice. The authors should provide either a theoretical justification or empirical evidence supporting the necessity of both.
- Eq5 is hard to follow, can you provide a more detail explain why the mutal information can approximate by edundant (R), unique (Um), and synergistic (S).
- What's the definition of $Dist$ of $Sep$ in eq6?
- The comparison baseline across different modalities is not convincing. The baseline presented in Table 1 is too simplistic; stronger baselines should be used for a fairer comparison. For mass spectrometry, you could consider Spec2Mol [1] and DiffMS [2]; for NMR, NMR2Struct [3]; and for IR, Spectra-to-Structure [4]. I would also like to point out that the results in the dataset paper [5] have been updated—the top-1 accuracy for elucidation using NMR is now 73.38. In this context, the performance improvement achieved by MM-Spectrum appears modest.

[1] An end-to-end deep learning framework for translating mass spectra to de-novo molecules[J]. Communications Chemistry, 2023, 6(1): 132.

[2] Diffms: Diffusion generation of molecules conditioned on mass spectra[J]. arXiv preprint arXiv:2502.09571, 2025.

[3] Accurate and efficient structure elucidation from routine one-dimensional NMR spectra using multitask machine learning[J]. ACS Central Science, 2024, 10(11): 2162-2170.

[4] Spectra to structure: contrastive learning framework for library ranking and generating molecular structures for infrared spectra[J]. Digital Discovery, 2024, 3(12): 2417-2423.

[5] Unraveling molecular structure: A multimodal spectroscopic dataset for chemistry. Advances in Neural Information Processing Systems, 37:125780–125808, 2024

- The major concern is that the evaluation is limited to a simulated dataset, offering little insight into real-world applicability.

---

> ### Author Rebuttal · Authors · 2026-03-31
>
> Thank you for the feedback and spending your time reviewing our paper. Please find below responses to your comments and the changes we will make to the paper.
>
> > # R1. Clarity of token definitions and semantic density (Q1 & Q2)
>
> We appreciate your attention to this detail. In the paper, inputs are not raw continuous signals passed directly into the Transformer. Instead, each modality is first converted into a structured sequence representation suitable for sequence modeling. In the appendix B.5, we have clarified the tokenization for each modality.
>
> *Table 1: Example tokenization / serialization*
>
> |Modality|Raw Signal Form|Tokenization|Example Token Sequence|
> | -------- | ------------------------------ | ------------------------------------ | -------------------------------- |
> |NMR|Chemical shifts, multiplets|Shift + symbol / multiplicity tokens|`[NMR] 7.26 s 7.18 d 6.91 d ...`|
> |IR|Continuous spectrum| Binned frequency intervals| `[IR] 2 7 12 0 8 3 ...`|
> |MS|Peak list (\(m/z\), intensity)|Filtered peak tokens| `[MS] 91.05 100 119.08 42 ...`|
>
> We also appreciate the reviewer’s concern regarding the notion of **semantic density**. Our intent is not to claim that MS lacks critical information—indeed, high-precision \(m/z\) values are very important in mass spectrometry. Rather, it is used in the paper as a relative modeling concept: under a shared sequence-modeling framework, different modalities induce token streams with different average structural constraint strength per token and different redundancy statistics.
>
>
> > # R2. Distinction between $e_m$ and $b_m$ (Q3)
>
> We thank the reviewer for this observation. In our design, they serve different roles:
>
> - $e_m$: representation-level identity injection, i.e., a learnable modality identity signal appended to the token representation;
> - $b_m$: routing-level prior shift, i.e., a modality prior that stabilizes coarse routing boundaries in the router’s decision space.
>
> Geometrically, $e_m$ changes the representation itself, whereas $b_m$ acts more like a translation term in the router’s decision boundary. This separation is helpful because it stabilizes early-stage modality-aware routing without removing the role of content $h_i$ in later fine-grained expert matching.
>
> In the paper, **Table 4 (routing ablation)** already shows that removing either component degrades performance, and removing both degrades it further.
>
> > # R3. Clarification of Eq. (5), Eq. (6), *Dist*, and *Sep* (Q4 & Q5)
>
> Thank you for pointing this out. Our use of PID is intended as an architectural inductive bias, not as a strict information-theoretic estimation procedure. Specifically, the expression is used to motivate why the expert pool is organized into:
> - shared experts,
> - modality-specific experts,
> - interaction experts.
>
> That is, the role of PID here is to provide a principled structural view of *what kinds of information pathways should be separated*,  rather than being a quantity explicitly optimized or recovered.
>
> We also agree that *Dist* and *Sep* should be explicitly defined. In the revision, we will clarify that they are trainable surrogates encouraging the desired expert behavior—namely, redundant-consistent shared experts and unique-separable modality-specific experts.
>
> A concise form is:
>
> ${Dist}(o_i,o_j)=\|o_i-o_j\|_2^2$
>
> ${Sep}(\mu_m,\mu_{\neg m})=\max(0,\gamma-d(\mu_m,\mu_{\neg m})).$
>
> Here,
>
> - *Dist*: an $L_2$ consistency term encouraging shared experts to produce similar representations across modalities;
> - *Sep*: a margin-based separation term encouraging modality-specific experts to remain distinguishable from the others.
>
> > # R4. Comparison with specialized single-modality models (Q6)
>
> We appreciate this suggestion and agree that works such as Spec2Mol, DiffMS, NMR2Struct, and Spectra-to-Structure are important specialized baselines in their respective modalities. We will expand the related work to discuss them more carefully.
>
> At the same time, we would like to clarify the scope of our paper. MM-Spectrum is not intended as a modality-specific solver specialized for only MS, only NMR, or only IR. Rather, it is a general multispectral fusion framework aimed at resolving the specific failure mode of full-modality collapse.
>
> This distinction is important for interpreting the contribution. The practical motivation in real chemistry settings is precisely to better leverage multiple spectra jointly, not to replace every modality-specific solver.
>
> > # R5. Real-world applicability beyond simulated data (Q7)
>
> Thank you for pointing this out, we have added experiments on SDBS and an MMST data source. Preliminary evidence indicates that MM-Spectrum maintains the same qualitative advantage across both simulated and real-data settings.
>
> *Table 2: Cross-dataset generalization results*
>
> |Dataset|Setting|Top-1%|Top-10& |
> |------- |--------|:-------:|:-------:|
> |SDBS|Baseline|37.56|59.84|
> |SDBS|Ours|**59.52**|**76.32**|
> |MMST|Baseline |65.71|89.02|
> |MMST|Ours| **87.35**|**98.72**|

---

> > ### Author Rebuttal · Reviewer_z3h6 · 2026-04-02
> >
> > I thank the author for the clarification. However, I am not convinced by the argument that MM-Spectrum is not intended as a modality-specific solver, given that you claim it is proposed to better leverage multiple spectra jointly. Yet two critical issues remain: (1) in practice, multi-spectrum data is often incomplete; and (2) if multi-spectrum performance proves inferior to single-spectrum approaches, what is the rationale for using it at all?
> >
> > More experimental details on SDBS and MMST should also be provided.

---

> > > ### Author Response · Authors · 2026-04-03
> > >
> > > We sincerely thank the reviewer for the careful follow-up and for pushing us to clarify the practical positioning of MM-Spectrum more precisely. Below we respond to each point in turn.
> > >
> > > > # R1. In practice, multispectral data is often incomplete
> > >
> > > We thank the reviewer for this valuable observation.
> > >
> > > The current submitted manuscript already contains direct evidence for this point in `Table 2` (in the submitted manuscript). Across all missingness settings, including balanced missingness and severe single-modality degradation, MM-Spectrum remains consistently and substantially stronger than the baselines.
> > >
> > > We further added a more explicit evaluation protocol: Train with *full-modality inputs*, then completely *remove one modality* at test time.
> > >
> > > *Table 1. Evaluation with one missing modality*
> > > |Missing Modality|Setting|Top-1|Top-5|Top-10|
> > > |----------------|--------|--------|--------|--------|
> > > |MS removed|Baseline|25.71|28.52|30.84|
> > > |MS removed|Ours|**73.94**| **86.12**|**88.47**|
> > > |IR removed|Baseline|31.52 |35.78|37.79|
> > > |IR removed|Ours| **74.81**|**86.79**|**89.05**|
> > > |NMR removed| Baseline|18.65|20.46|23.17|
> > > |NMR removed| Ours|**39.24**| **59.36** |**63.41**|
> > >
> > > These results show that even when modalities are partially missing, MM-Spectrum still maintains **a large and consistent margin** over the baseline.
> > >
> > > > # R2. Why multispectral modeling still matters
> > >
> > > We appreciate the reviewer’s constructive suggestion on how to strengthen this section.
> > >
> > > This is exactly the phenomenon studied in `Table 1` of the submitted paper. MM-Spectrum does satisfy the most important practical criterion: the full-modality MM-Spectrum model is already stronger than all of its own single-modality baselines.
> > >
> > > - The dense baseline suffers a severe collapse from *69.83%* Top-1 to *44.29%* Top-1 (full-modality);
> > > - By contrast, MM-Spectrum not only avoids collapse, but turns full-modality into the best-performing setting.
> > >
> > > This point is also reinforced by the paper’s `Table 3` (HAC-stratified evaluation). As molecular complexity increases, the improvement of MM-Spectrum over the baseline becomes *larger*, not *smaller*.
> > >
> > > In real-world structural elucidation, NMR, IR, and MS provide constraints at different structural granularities.
> > >
> > > A representative example from the chemical informatics literature demonstrates how algorithms can mimic the spectroscopist-driven process by simultaneously utilizing IR, NMR, and MS data:
> > > - "Database independent automated structure elucidation of organic molecules based on IR, 1H NMR, 13C NMR, and MS data." *Journal of chemical information and modeling* 61.2 (2020)
> > > - "Spectrometric identification of organic compounds." *Journal of Chemical Education* 39.11 (1962).
> > >
> > > These references make clear that in chemical practice, no single modality is universally sufficient, and multispectral evidence is often necessary precisely.
> > >
> > > > # R3. More experimental details on SDBS and MMST
> > >
> > > We thank the reviewer for the valuable feedback. In the revised paper, we will provide a fuller description.
> > > - `SDBS` is a real-world experimental spectral dataset collected from the official SDBS database;
> > > - `MMST` is used as a supplementary large-scale multispectral benchmark.
> > >    - *Source*:  Advancing Structure Elucidation with a Flexible MultiSpectral AI Model. *Angewandte Chemie* 138.2 (2026).
> > >
> > >  *Table 2. Additional datasets used in the rebuttal*
> > >
> > > |Dataset|Type|TotalSamples|Train|Val|
> > > |-------|----------|------------|--------|------|
> > > |SDBS|Real-World|9,289|7,942|418|929|
> > > |MMST|Simulation|5,137,355|4,510,368|563,796|63,191|
> > >
> > > Regarding implementation details, for SDBS dataset, we used a more conservative optimization setting, including reduced effective batch size from 8192 to 2048 and stronger regularization by increasing dropout from 0.10 to 0.20.
> > >
> > > On these two additional data sources, we conducted experiments under multiple settings.
> > >
> > > *Table 3. Cross-dataset evaluation protocol*
> > >
> > > |Dataset| Training Protocol| Modality Setting|Method|Top-1 |Top-5 |Top-10|
> > > |-------|-------------------|----------------|-----------|--------|--------|--------|
> > > |SDBS|From Scratch|NMR|Baseline|18.15|31.74|33.69|
> > > |SDBS|From Scratch|MS|Baseline|5.38|11.07|13.92|
> > > |SDBS|From Scratch|IR|Baseline|13.21|25.48|26.61|
> > > |SDBS|From Scratch|Full| Baseline|14.10|27.13|31.53|
> > > |SDBS|From Scratch|Full| Ours|**26.67**|**38.91**| **47.37**|
> > > |SDBS|Pretrain+Finetune|Full|Baseline|37.56 |57.15 |59.84|
> > > |SDBS|Pretrain+Finetune|Full|Ours| **59.52**|**73.95**|**76.32**|
> > > |MMST|From Scratch|NMR|Baseline|79.41|89.45|94.71|
> > > |MMST|From Scratch|MS|Baseline|25.13|28.42|31.28|
> > > |MMST|From Scratch|IR|Baseline|19.35|23.60|27.04|
> > > |MMST|From Scratch|Full|Baseline|65.71|85.07|89.02|
> > > |MMST|From Scratch|Full|Ours|**87.35**|**95.87**|**98.72**|
> > >
> > > We appreciate the reviewer’s feedback and valuable guidance.
> > >
> > > ---
> > >
> > > If our answers have justifiably addressed your concerns, we respectfully hope that you could increase your score to support the acceptance of our work. Look forward to your further reply!

---

### Official Review · Reviewer_WU52 · 2026-03-13

**Soundness:** 3
**Presentation:** 3
**Significance:** 3
**Originality:** 2
**Overall Recommendation:** 5
**Confidence:** 4

**Summary:**

This paper studies a real and well-motivated problem in multimodal molecular structure elucidation: adding more spectra does not automatically help, and in fact naive full-modality fusion can hurt badly. The authors propose a simple but effective improvement in the fusion layer through a modality-aware MoE design, where experts are organized to capture shared, modality-specific, and cross-modal interaction information. The main empirical result is strong: on the same benchmark used by prior works, the proposed model turns the full-modality failure of dense concatenation into a clear improvement, reaching 76.04% Top-1 versus 44.29% for the baseline, while also improving bimodal and missing-modality settings. I think the paper’s main contribution is therefore less about a new chemistry dataset or prediction target, and more about showing that a better multimodal mixing mechanism can materially improve spectra-to-structure models.

**Compliance With Llm Reviewing Policy:**

Affirmed.

**Final Justification:**

upgraded the score.

**Key Questions For Authors:**

1. Generalization to experimental spectra
All experiments appear to use the Alberts et al. benchmark. Have the authors evaluated MM-Spectrum on other spectral datasets or experimentally collected spectra (e.g., SDBS or similar libraries)? Such experiments could help assess the robustness of the proposed fusion mechanism to real-world measurement noise and instrument variability.

2. Comparison with other multimodal fusion strategies
The paper focuses on improving multimodal fusion through a structured MoE design. However, the experimental comparison only includes the dense concatenation baseline from Alberts et al. (2024). Could the authors compare against other multimodal learning strategies already cited in the paper, such as PCGrad (Yu et al., 2020), OGM-GE (Wu et al., 2022), or contrastive alignment approaches (Liang et al., 2022)? This would help isolate whether the gains arise from the MoE design specifically or from improved multimodal training strategies more broadly.

3. Computational cost vs dense models
MoE architectures often introduce routing overhead and load balancing complexity. Could the authors report training and inference cost relative to the dense baseline (e.g., FLOPs, wall-clock training time, or memory usage)? This would help evaluate the practical benefits of the proposed sparse routing.

**Limitations:**

yes

**Strengths And Weaknesses:**

Strengths

1. Clear problem identification and motivation.
The paper identifies an important issue in multimodal molecular structure elucidation: incorporating additional spectral modalities can degrade performance when naive fusion strategies are used. The authors clearly demonstrate this failure mode in the baseline (Top-1 accuracy drops from 69.83% with NMR alone to 44.29% with full modalities), establishing a well-defined problem where improved multimodal fusion mechanisms could have significant impact.

2. Simple and well-motivated architectural improvement.
The proposed fusion strategy using a structured Mixture-of-Experts layer is conceptually simple yet well motivated. The decomposition of experts into modality-specific, shared, and interaction components inspired by the Partial Information Decomposition framework is a reasonable design choice and provides an interpretable lens for multimodal learning.

3. Strong empirical improvements on the benchmark.
The proposed model shows consistent improvements over the Alberts et al. (2024) baseline across multiple settings, including full-modality, bimodal, and missing-modality scenarios. The gains reported in Table 1 are substantial, suggesting that the fusion mechanism meaningfully improves the ability of the model to combine heterogeneous spectral signals.

4. Informative ablations.
The ablation experiments in Tables 4–6 are useful and help isolate the contribution of the routing signals and heterogeneous expert capacities. These experiments provide insight into which modalities the model relies on and how the MoE routing contributes to performance.

Weaknesses
1. Limited baseline comparisons for a fusion architecture paper.
The proposed method is only compared against the dense concatenation baseline of Alberts et al. (2024). Given that the main contribution of the work is a new multimodal fusion strategy, it would be important to compare against other multimodal optimization or fusion techniques that are already cited in the paper, such as PCGrad (Yu et al., 2020), OGM-GE (Wu et al., 2022), or contrastive alignment approaches (Liang et al., 2022). Without these comparisons, it is difficult to determine whether the reported improvements arise from the MoE design itself or from adopting a more appropriate fusion strategy in general.

2. Assumptions behind heterogeneous expert capacities are not validated.
The heterogeneous expert design assumes that NMR information requires larger experts (2048-dimensional) while IR and MS experts can be smaller (512-dimensional). While the authors attribute this to chemical prior knowledge, the justification remains somewhat speculative. An ablation demonstrating that larger NMR experts improve performance would make this design choice more convincing.

3. Interpretation of the reported improvement is unclear.
The reported 31.75% improvement over the baseline is significant, but because only a single baseline is evaluated, it is difficult to disentangle how much of the gain comes from the MoE architecture versus simply avoiding dense concatenation.

---

> ### Author Rebuttal · Authors · 2026-03-31
>
> Thank you for the feedback and spending your time reviewing our paper. Please find below responses to your comments and the changes we will make to the paper.
>
> > # R1. Limited comparisons to other fusion baselines (W1&Q2)
>
> We would like to thank the reivewer for this valuable suggestion. We first aligned with the canonical dense concatenation baseline of Alberts et al. (2024), because the central failure mode studied in this paper—*full-modality collapse*—is most clearly exposed in that setting. Furthermore, we also added comparisons against several representative alternatives.
>
> *Table 1: Comparison with other fusion baselines*
>
> |Method|Top-1%|Top-5%|Top-10%|
> | -----------------------------------------|:-------:|:-------:|:-------:|
> |Dense Concatenation |44.29|59.39|62.04|
> |Cross-Attention Fusion|42.15|56.45 |61.91|
> |Contrastive Alignment (Liang et al., 2022) |39.01|52.74|57.06|
> |OGM-GE (Wu et al., 2022) |44.87| 60.93 |63.32|
> |PCGrad (Yu et al., 2020) |44.73 |61.24| 64.92|
> |Gradient Blending (Wang et al., 2020) | 46.78|63.45|67.82|
> | **MM-Spectrum (ours)** | **76.04**| **87.83**|**90.26**|
>
> Our observation is that some optimization-level methods do partially alleviate the degradation of the dense baseline, but their improvements remain clearly below MM-Spectrum.
>
> This suggests that generic gradient modulation or representation alignment methods alone are insufficient to fundamentally resolve the multimodal imbalance induced by multispectral heterogeneity. By contrast, MM-Spectrum does not only mitigate conflicts at the optimization level; it also structurally disentangles redundant, unique, and synergistic information pathways through expert organization.
>
>
> > # R2. Generalization to additional datasets (W3&Q1)
>
> We are thankful for the practical recommendation, which has helped improve the robustness of our method. We have performed additional experiments on two extra data sources:
>
> 1. `SDBS`, a real-world experimental spectral dataset;
> 2. `MMST`, an MultiModalSpectralTransformer-related source [1].
>
> *Table 2: Additional datasets and scale*
>
> |Dataset|DataType|TotalSamples|Train|Val|Test|Modalities|
> |:------|:---------|:-----------:|:-----:|:----:|:---:|:---------------------------------:|
> |**SDBS**|Real-world|9289|7942|418|929|IR/MS/1H-NMR/13C-NMR|
> |**MMST**|Simulation|5997971|5275360|659420|63191|IR/1H-NMR/13C-NMR/COSY/HSQC|
>
> Importantly, we did not only test different datasets. We also considered multiple training protocols.
>
> *Table 3: Cross-dataset generalization results*
>
> |Dataset|TrainingProtocol|Setting|Top-1%|Top-5%|Top-10%|
> |-------|-------------------|--------|:-------:|:-------:|:-------:|
> |SDBS|FromScratch|Baseline|14.10|27.13|31.53|
> |SDBS|FromScratch|Ours|**26.67**|**38,91**|**47.37**|
> |SDBS|Pretrain&Finetune|Baseline|37.56|57.15|59.84|
> |SDBS|Pretrain&Finetune|Ours|**59.52**|**73.95**|**76.32**|
> |MMST|FromScratch|Baseline|65.71|85.07|89.02|
> |MMST|FromScratch|Ours|**87.35**|**95.87**|**98.72**|
>
> Our observation is that MM-Spectrum shows the same qualitative trend across datasets and protocols, suggesting that the method is not merely exploiting a benchmark-specific artifact, but addressing a more general structural challenge in multispectral fusion.
>
> [1] Priessner, Martin, et al. "Advancing Structure Elucidation with a Flexible Multi‐Spectral AI Model." Angewandte Chemie 138.2 (2026): e17611.
>
> > # R3. Validation of heterogeneous expert capacities (W2)
>
> We appreciate your attention to this detail. From a science perspective, NMR tokens tend to require more nonlinear capacity to preserve and transform subtle local structure cues, while IR/MS tokens often benefit more from selective filtering and lighter processing.
> This is precisely why we propose capacity–utility matching: experts with larger nonlinear capacity should preferentially serve higher-value, more constraint-rich tokens.
>
> To further validate this assumption, following your suggestion, we added a direct capacity ablation study.
>
> *Table 4: NMR expert capacity ablation*
>
> |NMRExpertDim|Top-1%|Top-5%|Top-10%|
> |:------------:|:-------:|:-------:|:-------:|
> |512|73.81|85.69|87.04|
> |1024|74.65|86.27|88.63|
> |2048|**76.13**|87.94|**91.06**|
> |4096|76.09|**88.02**|90.87|
>
> > #  R4. Computational efficiency (Q3)
>
> We agree that, for a sparse-routing model, performance should be accompanied by a clear discussion of cost–benefit trade-offs. We would like to clarify that the current submission already includes a detailed computational analysis in the appendix—specifically in **Tables 10, 11, and 12**.
>
> More concretely:
>
> - *Table 10* analyzes the effect of expert number and Top-k, showing that the final configuration is chosen not only for accuracy but also for a favorable efficiency trade-off.
> - *Table 11* reports the sensitivity of compute and performance to routing sparsity, supporting our design choice.
> - *Table 12* compares resource usage and shows that MM-Spectrum achieves a much better performance–cost Pareto trade-off than the dense baseline.

---

> > ### Author Rebuttal · Reviewer_WU52 · 2026-04-05
> >
> > Thank you for addressing all my comments and doing additional experiments.

---

> > > ### Author Response · Authors · 2026-04-06
> > >
> > > Thank you very much for your thoughtful follow-up and for taking the time to read our additional responses and experiments.
> > >
> > > We are grateful that our rebuttal and additional experiments were able to address your concerns. Your comments and suggestions have been very valuable in helping us improve the quality of this work. We will carefully incorporate your valuable suggestions and the corresponding clarifications into the revised version of the paper.
> > >
> > > Thank you again for your support and for your invaluable contribution to our research.

---

### Decision · Program_Chairs · 2026-04-30

**Decision:**

Accept (regular)

**Comment:**

After considering the reviews and rebuttal, I support acceptance / weak acceptance. The paper addresses a meaningful and well-motivated problem in multimodal molecular structure elucidation: naive fusion of additional spectral modalities can actually hurt performance. Across the reviews, there is broad agreement that this failure mode is interesting and practically important, and that the proposed modality-aware heterogeneous MoE is a technically sensible response. The empirical improvements over the dense concatenation baseline are large, and the rebuttal materially strengthened the paper by adding comparisons to stronger multimodal baselines, additional experiments on SDBS and MMST, and further missing-modality results. Two reviewers explicitly stated that their concerns were fully resolved after rebuttal, while another remained positive throughout.

The main unresolved concern is the lack of direct comparison to specialized single-modality methods, which one reviewer viewed as important for judging practical usefulness. I agree this is a real limitation, and it means the paper should not be overclaimed as the strongest structure-elucidation system overall. However, I do not think this undermines the paper’s central contribution as a multimodal fusion paper: the evidence supports the claim that the proposed architecture mitigates the full-modality collapse of naive fusion and improves performance across full-modality, bimodal, and missing-modality settings. The remaining issues around presentation, notation, and positioning seem revision-level rather than fundamental soundness problems, so on balance I would recommend acceptance.